# ConforFormer: representation for molecules through understanding of conformers

## Abstract

Recent years have seen a growing interest in machine learning approaches for chemical tasks. The best existing methods focus on building base models that combine molecular graphs ("2D structures") with atomic coordinates in 3D to predict molecular properties, typically through pre-training followed by fine-tuning on benchmark datasets. However, current approaches require retraining the entire model for each prediction task, using published weights only as initialization. While this enables state-of-the-art performance, it limits practical deployment, as real-world datasets are often too small to support the stable retraining of large models. Importantly, the 3D geometry of a molecule holds crucial information for predicting its properties, but a single molecular graph usually corresponds to several 3D geometries, called conformers, introducing ambiguity into the inference process. Typical solutions rely on molecular graphs, but this approach is not easily generalizable beyond organic molecules. Here, we present Confor-Former, a method that explicitly accounts for the diversity of 3D conformations of a molecule to derive a task-agnostic and conformation-agnostic vector representation. This model serves as a foundational framework, producing embeddings that can be generated once and directly applied to downstream tasks, including property prediction and structural similarity, without extensive fine-tuning.

## 1 Introduction

Pre-training large foundational models via self-supervised learning has become a major focus of modern representation learning. The remarkable success of such architectures in text and vision tasks has inspired applications in the natural sciences, including chemistry (Choi et al., 2025), physics (Wiesner et al., 2025), and applied meteorology (Bodnar et al., 2025).

Existing pre-trained chemical models are typically used to initialize weights for supervised prediction tasks (Zhou et al., 2022; Ahmad et al., 2022; Wang et al., 2022; Chithrananda et al., 2020; Fang et al., 2022) While this approach can achieve state-of-the-art performance, it is often unstable on real-world chemical datasets, which in laboratory settings rarely exceed a few hundred experimentally measured points (Ahneman et al., 2018).

Most chemical foundation models still operate on simplified 2D representations of molecules, ignoring the conformation and configurational diversity that governs their real chemical behavior. In reality, each compound exists as an ensemble of 3D structures (conformers) whose distribution determines such properties as binding affinities, docking poses, and chemical reactivity (Kuznetsov et al., 2024; Laplaza et al., 2024; Finta et al., 2025). Typically, conformers differ from each other by rotations around single bonds, inversion of nitrogen lone pairs and other movements allowed by molecular flexibility. Conformers are distinct from isomers, which also are 3D geometries with the same composition, but one isomer cannot be produced from another without rearranging chemical bonds, i.e. a chemical reaction happening. Capturing the distribution of 3D geometries possible for a molecule is essential for the property prediction task, yet we are not aware of approaches where understanding of conformations is explicitly incorporated as a learning task in foundation models.

Contrastive learning has emerged as a powerful strategy to enhance foundation models and refine embeddings without explicit labels by regularizing the embedding space in a way that it becomes organized so that distance correlates with semantic similarity. By structuring the embedding space

to bring similar objects closer while pushing dissimilar ones apart, models learn more informative, general-purpose representations. Methods developed at Amazon (Jiang et al., 2023; Ak et al., 2025) illustrate how contrastive approaches can refine embeddings across modalities, improving downstream task performance. A notable example is Microsoft E5 (Wang et al., 2024), trained in a weakly supervised manner on naturally occurring document pairs such as questions and answers from forums.

To our knowledge, no chemical embedding model capturing the diversity of 3D molecular conformations has yet been published. Here we introduce ConforFormer, a foundational model that explicitly accounts for this diversity by aligning embeddings across multiple conformations of a molecule to produce compact, informative representations suitable for downstream tasks. In this work, we present 1) a new weakly supervised contrastive learning objective for molecular representations, 2) a benchmark evaluating the model's ability to distinguish pharmaceutically relevant molecules, and 3) the performance of the resulting embeddings on established chemical benchmarks. The training code, inference examples, and model weights for generating these embeddings are publicly available under MIT or CC-BY licenses.

## 2 Preliminaries

### 2.1 Backbone Models for Molecular Embedding

Backbone architectures for molecular embeddings generally fall into three categories. Graph-based models such as MPNNs (Yang et al., 2019) and Grover (Rong et al., 2020) represent molecules as atom-bond graphs and capture local connectivity through message passing. While effective for property prediction, their exclusive reliance on 2D topology limits their representational ability. Sequence-based transformers adapt NLP methods to SMILES (Weininger, 1988) strings. Models like MolBERT (Li & Jiang, 2021), ChemBERTa (Chithrananda et al., 2020) and ChemBERTa-2 (Ahmad et al., 2022) reuse the strength of the text-based transformers while treating individual atoms in the structure-encoding string as tokens. These models are easy to train on huge datasets but they suffer from the same limitations as graph-based models. To address these issues, 3D-aware models have been developed. Methods such as GEM (Fang et al., 2022) and ABT-MPNN (Liu et al., 2023) augment 2D graphs by 3D structural info, improving accuracy on tasks that depend on spatial structure. Among these, Uni-Mol family of models (Zhou et al., 2022; Lu et al., 2024; Ji et al., 2024) has established itself as one of the leading frameworks. Built on a transformer backbone with explicit 3D positions encoding it achieves state-of-the-art performance in property prediction, conformation generation, and docking.

### 2.2 Uni-Mol Architecture

Uni-Mol employs an E(3)-equivariant transformer (the distinction between SE(3) and E(3) is discussed in Dumitrescu et al. (2024)). Each atom is represented as a token embedding that incorporates its element type as a categorical feature. Spatial information is encoded through pairwise distance matrix representation, which is integrated into the attention mechanism of the multi-layer, multi-head transformer encoder as an initial attention mask. The model is pretrained on a dataset introduced in the same paper which contains 19M SMILES using self-supervised tasks such as 3D position recovery and masked atom prediction, allowing it to learn both chemical connectivity and 3D geometric relationships. Further developments in Uni-Mol family of methods focuses on improving quantum-chemical property prediction on selected benchmarks. Uni-Mol+ (Lu et al., 2024) adds molecular graph and richer atom features, it also augments input geometry by adding inexpensive 3D geometry data points obtained from energy minimization trajectories. Following that work, Uni-Mol2 (Ji et al., 2024) introduces larger models and explores scaling laws up to a 1B parameter model. However, these later models introduce reliance on 2D connectivity which in our opinion limits the generalization ability of the method, a challenge addressed by ConforFormer and discussed in the subsequent section.

## 2.3 TECHNICAL PRACTICES IN TRANSFER LEARNING WITH FOUNDATIONAL TRANSFORMER MODELS

In practice, large transformer models pretrained with self-supervised objectives (e.g., masked token prediction, contrastive learning, or distance-based representations) are rarely fully retrained for downstream benchmarks. Instead, most approaches leverage transfer learning by freezing the majority of pretrained layers and fine-tuning only a small subset, such as the top layers or task-specific heads. This strategy significantly reduces computational cost while retaining the rich representations learned during pretraining. For example, in NLP, models like BERT (Devlin et al., 2019) are frequently fine-tuned using adapter modules or by unfreezing just the top 2–4 transformer layers (Houlsby et al., 2019) achieving strong performance on GLUE (Wang et al., 2018) or Super-GLUE (Sarlin et al., 2020) with minimal gradient updates. These practices (e.g., layer freezing and lightweight task-specific adapters) allow researchers to exploit the foundational knowledge captured during self-supervised pretraining while keeping downstream training efficient and scalable.

## 2.4 CHEMICAL PERSPECTIVE FOR MOLECULAR REPRESENTATIONS

Most of the common benchmarks for molecular property prediction are structured as lists of SMILES strings with target values provided. This framing of the task is natural for the dataset builders, and it is unsurprising that nearly every model for prediction of molecular properties treats the existence of molecular graph as a given fact and the 2D structure (technically "structural formula") as the underlying object which can be augmented by features like 3D geometry. However, from a physical point of view, molecular graphs do not exist. Molecules are flexible 3D objects, and each molecule can be represented by an area on the potential energy surface corresponding to a range of geometries. Distinct molecules are such areas surrounded by a high enough potential energy barrier so that they don't interconvert. The structural formulas usually denote one area each. The points belonging to the same area are called "conformers": different 3D geometries possible for the same molecule.

So, structural formulas (molecular graphs, 2D structures) are in practice just a labeling scheme designed to distinguish chemically distinct molecular compounds and feature engineered for readability by chemists. The structural formulas work well for the chemistry of organic molecules, but for more complex compounds, particularly organometallics, there is no agreed-upon approach to represent those as a molecular graph. There is an unmet need to develop a molecular representation which is not affected by the limitations and underlying assumptions of the molecular graph notation, particularly for the tasks of predicting the catalytic activity of metal-organic complexes (Bougueroua et al., 2025; Kalikadien et al., 2024).

## 3 CONFORFORMER: CONFORMER-BASED CONTRASTIVE LEARNING FOR MOLECULAR REPRESENTATIONS

Our approach builds upon the backbone of the Uni-Mol architecture with a novel contrastive learning objective to obtain performant, compact frozen embeddings for molecules which do not rely on a molecular graph for inference.

### 3.1 REPLICATING UNI-MOL AND REUSING IT FOR MOLECULAR EMBEDDINGS

Uni-Mol is one of the very few pretrained models for chemical representations which uses exclusively 3D coordinates and types of atoms as input. The details of the framework are outlined in the original work (Zhou et al., 2022). The Uni-Mol code, including both model weights and the training scripts, is available freely on GitHub. The respective dataset contains 10 conformers per each molecule, with 209M geometries corresponding to 20.9M unique SMILES strings. Since we aimed at using Uni-Mol as a transformer-based backbone of ConforFormer, we started with re-training the model from scratch using the provided code. It turned out that the model used an unconventional approach to supply the molecular graph information during training: 9% of the training dataset consisted of specially generated "flat" structures, where Z coordinates of atoms were set to zero, while X and Y denoted coordinates for drawing the molecule on a page or a monitor (after appropriate rescaling). These structures were not present in the published dataset but rather were generated on

the fly with RDKit from the SMILES string. So, we cannot guarantee that our replication was 1:1 corresponding to one used by Uni-Mol team since the RDKit feature to generate these geometries is currently deprecated and is not consistent between older versions. The Uni-Mol replicate lines in the tables denote our best attempt at retraining the model with "flattened" geometries included. We retrained the model without these structures with negligible degradation on the benchmarks (see Tables 1 and 2).

The fine-tuning process as described in the original Uni-Mol publication involved re-training the whole model initialized by the weights from the pre-training process. For completeness, we include the result of fine-tuning initialized from random weights; the results are surprisingly competitive, particularly on the QM9 benchmark, suggesting that at least some datasets contain enough structural information in the training set that a flexible chemistry-aware architecture can saturate these datasets without outside information.

The model architecture is Transformer-based, with tokens representing atoms. Following BERT, the Uni-Mol architecture has a "zero token" called CLS, with a special "empty" atom type and coordinates at the geometric center of the molecule. This token is processed as any of the atoms through the transformer but not included in the atom masking or distance prediction tasks, so it in practice holds information about the whole molecule. During finetuning it is extracted and passed through a dense MLP to the downstream tasks.

The model architecture throughout this whole document is that of the standard Uni-Mol up until contrastive learning is done. We tested the models on the MoleculeNet benchmarks (Wu et al., 2018). The training hyperparameters and the dataset and benchmark descriptions are provided in the sections A and C of the Supporting Information (SI).

## 3.2 TRANSFER LEARNING APPROACH ANALYSIS

To explore generalization ability of the model, we froze the pretrained model and kept the last 0, 1, and 3 layers for downstream tasks learning. In the case of the fully frozen model, all the training is in practice handled by the shallow MLP. Unsurprisingly, freezing the whole model (0 active layers) led to a noticeable decrease in performance (Tables 1 and 2), but the reproducibility of fine-tuning results visibly improved. We measured stability of the training by running the fine-tunings five times with different random seeds. The "Current work" section of the Tables 1 and 2 contains the mean and std values over these runs. As can be seen, for the chemically relevant classification benchmarks the standard deviation of the run results is 2-3x lower for the fully frozen models, suggesting a reasonable tradeoff between better learning capacity and the stability of the training results should be achievable. We should note that unfreezing a few layers of the model improved the performance in benchmarks to nearly fully unfrozen level (see section E, Ablation studies, in the Supporting information), so we see a potential for improvement here. There are strategies to construct adaptors on top of foundational models that improve the performance on downstream tasks (Hu et al., 2021). Optimizing the adaptor architecture goes beyond the scope of this study, so we kept the same MLPs with the same settings as in the Uni-Mol benchmarks for consistency.

## 3.3 CONTRASTIVE OBJECTIVE

Following the published studies in the text and image processing domains (Ak et al., 2025; Wang et al., 2024), the next step was to improve the frozen embedding performance with contrastive learning.

To make the resulting embedding stable to the propagated 3D representation of the molecule, we introduce a contrastive learning objective. We implement contrastive learning as a separate task within the pre-training. Model learns to distinguish pairs of conformers among various molecules in a weakly supervised manner where molecular graph is not supplied to the model directly but is only used to label pairs on the data generation stage. The training was done with the original batch extended to include 1 additional conformer for each molecule. Since we aimed for the model to learn to distinguish 3D structures in this way without directly ingesting molecular graph information, the "flat" structures were excluded from this training.

We used the NT-Xent loss function (Chen et al., 2020) to teach the model to put the embeddings of different 3D representations close in the embedding space. It is done to explicitly regularize the

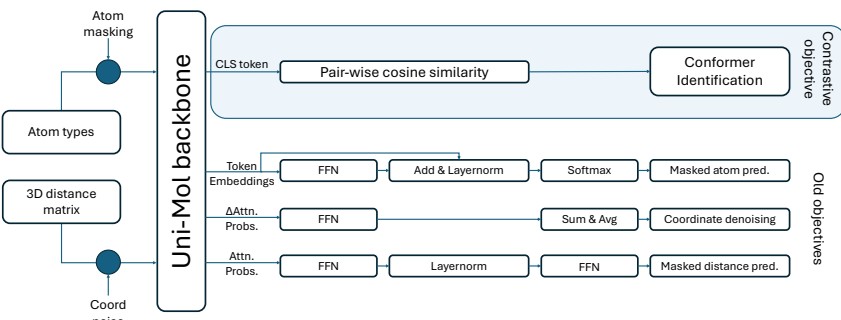

Figure 1: Schematic illustration of the ConforFormer framework: Model architecture with pretraining objectives.

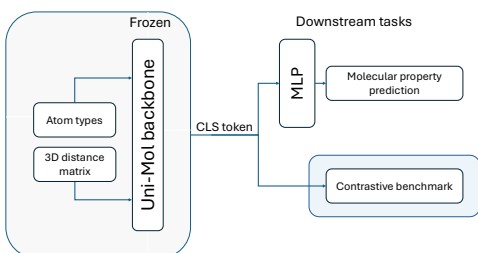

Figure 2: Schematic illustration of the ConforFormer framework: Finetuning scheme.

resulting embedding space so it has close embeddings for all the 3D representations of the molecule. This approach gives us the intended conformation-agnostic property.

Let $\mathbb{X}$ be the set of molecules and $f : \mathbb{X} \to \mathbb{R}^d$ an embedding function with $d = 512$ the size of embedding. For vectors $\boldsymbol{u}, \boldsymbol{v} \in \mathbb{R}^d$ we use a cosine-style similarity

$$\operatorname{sim}(\boldsymbol{u}, \boldsymbol{v}) \; := \; \frac{\boldsymbol{u}^\top \boldsymbol{v}}{\|\boldsymbol{u}\| \, \|\boldsymbol{v}\|} \; \in \; [0, 1] \quad \text{(by construction in our setup).}$$

In particular, for molecules $x, x' \in \mathbb{X}$ we write $\operatorname{sim}\big(f(x), f(x')\big)$, and we denote $\boldsymbol{z}_i := f(x_i)$.

In each training batch, we take $n = 128$ unique molecules and choose two 3D representations of each, yielding $2n$ embeddings $\{\boldsymbol{z}_i\}_{i\in\mathbb{B}}$ with index set $\mathbb{B} := \{1, \ldots, 2n\}$. Let $\mathbb{P} \subset \mathbb{B} \times \mathbb{B}$ be the set of ordered positive pairs, where $(i, j) \in \mathbb{P}$ iff $i$ and $j$ correspond to two conformers of the same molecule (with $i \neq j$). The NT-Xent contrastive loss is then defined as

$$\mathcal{L}_{\text{contrast}} \; := \; \sum_{(i,j)\in\mathbb{P}} -\log \frac{\exp\big(\operatorname{sim}(\boldsymbol{z}_i, \boldsymbol{z}_j)/\tau\big)}{\sum_{k\in\mathbb{B}\setminus\{i\}} \exp\big(\operatorname{sim}(\boldsymbol{z}_i, \boldsymbol{z}_k)/\tau\big)},$$

where $\tau > 0$ is the temperature. Higher $\tau$ reduces sensitivity to small embedding differences; in our experiments, we set $\tau = 0.07$ (see additional ablations in section E in the SI).

Overall, the total loss function ($\mathcal{L}_{\text{total}}$) for pre-training is as follows

$$\mathcal{L}_{\text{total}} = \mathcal{L}_{\text{token}} + 5 \cdot \mathcal{L}_{\text{coord}} + 10 \cdot \mathcal{L}_{\text{distance}} + 2 \cdot \mathcal{L}_{\text{contrast}}$$

Here $\mathcal{L}_{\text{token}}$ is the loss associated with the masked token prediction, $\mathcal{L}_{\text{coord}}$ is the loss associated with the coordinates de-noising task, and $\mathcal{L}_{\text{distance}}$ is the loss associated with the masked distance prediction. The batch for computing these losses (introduced in Zhou et al. (2022)) consists of $n = 128$ unique molecules unless otherwise specified (see section E in the SI for details).

This family of models trained with additional contrastive loss to distinguish conformers from other molecules better is further reported as ConforFormer in this study. Notably, training with the contrastive objective on the Uni-Mol dataset significantly improved the performance of the model on

Table 1: Biological activity (classification) benchmarks. Values denote ROC-AUC (higher is better). Literature data from the Uni-Mol paper.

| Model | BBBP | BACE | ClinTox | Tox21 | ToxCast | SIDER | HIV | MUV |
|---|---|---|---|---|---|---|---|---|
| *N points* | 2039 | 1513 | 1478 | 7831 | 8575 | 1427 | 41127 | 93087 |
| **Literature data** | | | | | | | | |
| D-MPNN | 0.710(3) | 0.809(6) | 0.906(6) | 0.759(7) | 0.655(3) | 0.570(7) | 0.771(5) | 0.786(14) |
| Attentive FP | 0.643(18) | 0.784(0) | 0.847(3) | 0.761(5) | 0.637(2) | 0.606(32) | 0.757(14) | 0.766(15) |
| N-GramRF | 0.697(6) | 0.779(15) | 0.775(40) | 0.743(4) | – | 0.668(7) | 0.772(1) | 0.769(7) |
| N-GramXGB | 0.691(8) | 0.791(13) | 0.875(27) | 0.758(9) | – | 0.655(7) | 0.787(4) | 0.748(2) |
| PretrainGNN | 0.687(13) | 0.845(7) | 0.726(15) | 0.781(6) | 0.657(6) | 0.627(8) | 0.799(7) | 0.813(21) |
| GROVER$_{base}$ | 0.700(1) | 0.826(7) | 0.812(30) | 0.743(1) | 0.654(4) | 0.648(6) | 0.625(9) | 0.673(18) |
| GROVER$_{large}$ | 0.695(1) | 0.810(14) | 0.762(37) | 0.735(1) | 0.653(5) | 0.654(1) | 0.682(11) | 0.673(18) |
| GraphMVP | 0.724(16) | 0.812(9) | 0.791(28) | 0.759(5) | 0.631(4) | 0.639(12) | 0.770(12) | 0.777(6) |
| MolCLR | 0.722(21) | 0.824(9) | 0.912(35) | 0.750(2) | – | 0.589(14) | 0.781(5) | 0.796(19) |
| GEM | 0.724(4) | 0.856(11) | 0.901(13) | 0.781(1) | 0.692(4) | 0.672(4) | 0.806(9) | 0.817(5) |
| Uni-Mol | 0.729(6) | 0.857(2) | 0.919(18) | 0.796(5) | 0.696(1) | 0.659(13) | 0.808(3) | 0.821(13) |
| **Current work** | | | | | | | | |
| *Unfrozen models* | | | | | | | | |
| Uni-Mol replicate | 0.705(26) | 0.832(25) | 0.857(22) | 0.788(3) | 0.685(7) | 0.644(14) | 0.784(8) | 0.784(9) |
| ConforFormer–OMol | 0.691(21) | 0.820(29) | 0.686(26) | 0.787(4) | 0.689(7) | 0.634(13) | 0.786(4) | 0.758(30) |
| Uni-Mol, no pretrain | 0.655(11) | 0.775(39) | 0.639(46) | 0.735(15) | 0.635(12) | 0.607(18) | 0.739(18) | 0.616(11) |
| *Frozen models* | | | | | | | | |
| Uni-Mol replicate | 0.640(4) | 0.775(2) | **0.767(10)** | 0.709(4) | **0.653(2)** | 0.606(13) | 0.734(3) | 0.755(10) |
| Uni-Mol no "flat" | 0.651(1) | 0.778(1) | 0.725(11) | 0.711(1) | 0.637(1) | 0.606(6) | 0.746(2) | 0.742(8) |
| Uni-Mol, OMol data | 0.664(4) | **0.783(4)** | 0.698(11) | 0.710(1) | 0.629(1) | 0.610(6) | **0.756(5)** | 0.695(7) |
| ConforFormer–UniMol | 0.665(7) | 0.731(55) | 0.533(14) | 0.753(1) | 0.644(3) | **0.649(2)** | 0.711(4) | 0.716(5) |
| ConforFormer–OMol | **0.673(6)** | 0.751(13) | 0.716(9) | **0.755(1)** | 0.638(3) | 0.640(6) | 0.751(4) | **0.774(6)** |

some of the benchmarks compared to Uni-Mol with no flat structures, with the most dramatic change ($0.019 \rightarrow 0.013$) on the challenging QM9 benchmark. The performance on the classification benchmarks, however, was unsatisfactory, with particularly low ROC-AUC on the ClinTox dataset.

### 3.4 TRAINING ON THE OPENMOLECULES DATASET

While the Uni-Mol dataset is large and contains a lot of conformations for each structure, the individual data points are of relatively low quality. They were obtained from the RDKit geometry generation process, which provides reasonable geometries but can deviate from the reality due to limitations of the method. Moreover, no analysis was performed to check how many distinct geometries are present among these 10 conformers for each molecule; there likely is a significant amount of duplication, particularly for the more rigid systems. Fortunately, FAIR has recently published a dataset of high-quality molecular geometries for training chemical models. That OpenMolecules dataset (Levine et al., 2025) contains a subset of molecules specifically for conformer analysis and generation, which could serve as a drop-in replacement for the Uni-Mol dataset in this study (see section C.2 of the Supporting Information for the data preparation details).

Training on this sub-dataset (further OMol) without contrastive loss produced results on par with Uni-Mol with no flat structures, but training with contrastive loss and freezing the model produced the embeddings of higher quality, performing the best on 4 out of 6 quantum-chemical benchmarks, and tying for the first place in 5 out of 8 classification benchmarks (the performance was lower than the frozen embedding from Uni-Mol for BACE, ClinTox and ToxCast). Notably, the overall performance of these frozen, stable embeddings with the labels extracted using a simple MLP was on par or exceeding most of 2020s level methods, except the fully unfrozen Uni-Mol and GEM, which use an order of magnitude more parameters and directly ingest both the molecular graph and the 3D geometry during fine-tuning and inference. This suggests that the dense 512-byte embedding learned to efficiently store and retrieve key properties of the molecular graph from the 3D geometry alone. Importantly, for evaluating the ConforFormer–OMol we used the geometries generated for the MoleculeNet benchmark by the Uni-Mol team, for consistency. We would expect better performance with better quality geometries, but this would be the focus of a future work.

Table 2: Quantum-chemical regression benchmarks (values denote RMSD, lower is better).

| Model | ESOL | FreeSolv | Lipo | QM7 | QM8 | QM9 |
|---|---|---|---|---|---|---|
| *N points* | 1128 | 642 | 4200 | 6830 | 21786 | 133885 |
| **Literature data** | | | | | | |
| D-MPNN | 1.05(1) | 2.08(8) | 0.683(16) | 103.5(86) | 0.0190(1) | 0.00814(1) |
| Attentive FP | 0.88(3) | 2.07(18) | 0.721(1) | 72.0(27) | 0.0179(10) | 0.00812(1) |
| N-GramRF | 1.07(11) | 2.69(8) | 0.812(28) | 92.8(40) | 0.0236(6) | 0.01037(16) |
| N-GramXGB | 1.08(8) | 5.06(74) | 2.072(30) | 81.9(19) | 0.0215(5) | 0.00964(31) |
| PretrainGNN | 1.10(1) | 2.76(0) | 0.739(3) | 113.2(6) | 0.0200(1) | 0.00922(4) |
| GROVER$_{base}$ | 0.98(9) | 2.18(5) | 0.817(8) | 94.5(38) | 0.0218(4) | 0.00984(55) |
| GROVER$_{large}$ | 0.90(2) | 2.27(5) | 0.823(10) | 92.0(9) | 0.0224(3) | 0.00986(25) |
| GraphMVP | 1.03(3) | – | 0.681(10) | – | – | – |
| MolCLR | 1.27(4) | 2.59(25) | 0.691(4) | 66.8(23) | 0.0178(3) | – |
| GEM | 0.80(3) | 1.88(9) | 0.660(8) | 58.9(8) | 0.0171(1) | 0.00746(1) |
| Uni-Mol | 0.79(3) | 1.48(5) | 0.603(10) | 41.8(2) | 0.0156(1) | 0.00467(4) |
| **Current work** | | | | | | |
| *Unfrozen models* | | | | | | |
| Uni-Mol replicate | 0.83(3) | 1.80(11) | 0.608(9) | 58.8(30) | 0.0160(1) | 0.00520(0) |
| ConforFormer–OMol | 0.91(2) | 1.99(5) | 0.642(11) | 53.8(18) | 0.0159(0) | 0.00542(4) |
| Uni-Mol no pretrain | 0.98(5) | 2.49(23) | 0.787(22) | 83.6(156) | 0.0186(6) | 0.00618(7) |
| *Frozen models* | | | | | | |
| Uni-Mol replicate | 1.15(3) | **2.64(6)** | 0.916(4) | **82.6(44)** | 0.0264(5) | 0.0184(12) |
| Uni-Mol no "flat" | 1.23(3) | 2.92(4) | 0.935(4) | 88.5(32) | 0.0263(2) | 0.01910(19) |
| Uni-Mol, OMol data | 1.18(1) | 3.00(6) | 0.949(6) | 89.9(57) | 0.0274(2) | 0.0202(3) |
| ConforFormer–UniMol | 1.17(3) | 3.38(5) | 0.807(5) | 104.9(90) | 0.0223(2) | 0.01258(17) |
| ConforFormer–OMol | **1.12(2)** | 3.53(7) | **0.752(7)** | 99.9(112) | **0.0219(2)** | **0.01172(31)** |

## 4 TRANSFORMER-BASED EMBEDDINGS VECTOR SPACE ANALYSIS

As a result of a contrastive loss applied, we would expect the model to gain the ability to distinguish molecules better. Even if a molecular graph is only used to generate a sample of 3D geometries and distinct labels for the contrastive loss, the model should be able to learn which transformations of the molecular geometry are "allowed" under the constraint of structure remaining the same. However, we did not construct the training objective in a way to specifically distinguish conformers from isomers. So, in this section we explore the emergent behavior of the obtained embeddings in generalizing beyond the supplied conformations to distinguish between isomers.

### 4.1 ISOMER/CONFORMER DISTINGUISHING BENCHMARK

We introduce a new benchmark dataset PharmIsomer to validate the models' capability to distinguish between conformers and isomers and explore the resulting embedding space. We generated 10 conformers each for a sample of molecules from ZINC20 (Irwin et al., 2020) not overlapping with OMol or Uni-Mol datasets and constructed a set of 3.3B molecular pairs (either conformers or isomers) with a 80/10/10 train/test/validation split (see section B in SI for the details). The dataset contains four types of molecular pairs: backbone isomers where the molecules have a different bond order with the same composition (99.50% of all pairs); conformers (0.39%), optical isomers where molecules are mirror images of each other (0.05%); and diastereomers where the molecular topology is the same but the relative configuration of optical centers and/or double bonds is different (0.06%).

### 4.2 ISOMER SIMILARITY

As the initial step, we plotted the distributions of cosine similarity densities for the embeddings obtained from Uni-Mol replicate and from the ConforFormer models (Figure 3). Interestingly, the CLS token directly from our replication of the Uni-Mol already showed some level of separation between conformers and isomers, with cosine similarity of embeddings for conformers being closer

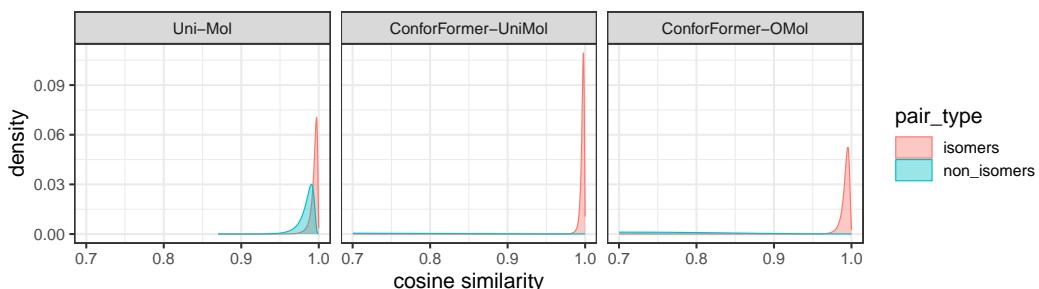

Figure 3: Distribution of cosine similarities between CLS token values extracted from Uni-Mol, ConforFormer–UniMol and ConforFormer–OMol, as measured on the PharmIsomer benchmark

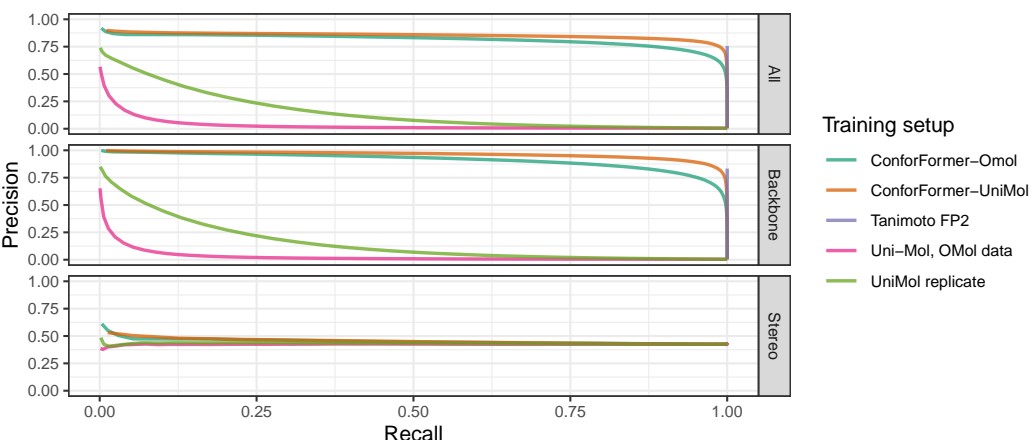

Figure 4: Precision and recall curves for different frozen representations on PharmIsomer benchmarks

to 1 than for isomers. This suggested from the start that a correctly trained model could learn to distinguish between those.

After including a contrastive objective, ConforFormer–Unimol and ConforFormer–OMol learn to cleanly separate conformers and isomers without any additional training. So, besides the embeddings becoming more useful for property prediction, they can be competitive for the tasks of similarity search as well. For that, we needed a more formal evaluation of the model capability to distinguish conformers and isomers.

Let $\mathbb{D} := \{(x_i, x_i', y_i)\}_{i=1}^N$ be a dataset of molecule pairs, where $x_i, x_i' \in \mathbb{X}$ and $y_i \in \{0, 1\}$ indicates the pair type: $y_i = 1$ for *conformers* and $y_i = 0$ for *isomers*. Define an index set $\mathbb{C} := \{i \in \{1, \ldots, N\} : y_i = 1\}$ with count $N_C := |\mathbb{C}|$.

Reusing the same similarity as in 3.3, define $s_i := \text{sim}\big(f(x_i), f(x_i')\big) \in [0, 1]$.

For a threshold $\theta \in [0, 1]$, predict *conformer* as $\hat{y}_i(\theta) := \mathbf{1}_{s_i \geq \theta}$ and define metrics as follows:

$$\text{Prec}(\theta) := \frac{\sum_{i \in \mathbb{C}} \mathbf{1}_{s_i \geq \theta}}{\sum_{i=1}^N \mathbf{1}_{s_i \geq \theta}}, \qquad \text{Rec}(\theta) := \frac{\sum_{i \in \mathbb{C}} \mathbf{1}_{s_i \geq \theta}}{N_C}.$$

The precision/recall curves constructed by sweeping over $\theta \in [0, 1]$ can be found on Figure 4. In this analysis, we treat enantiomers (mirror isomers) as the same molecule; the Uni-Mol backbone is based on a distance matrix, therefore has E(3) symmetry (Dumitrescu et al., 2024) and treats enantiomers as the same by design. For Uni-Mol replicate, the precision at 50% recall was just 8%;

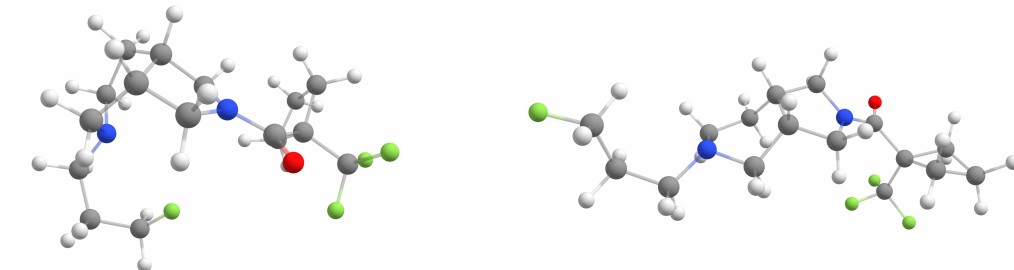

Figure 5: A pair of conformers of the same molecule having similarity of 0.93 in the Uni-Mol embedding space and 0.99 in ConforFormer–OMol

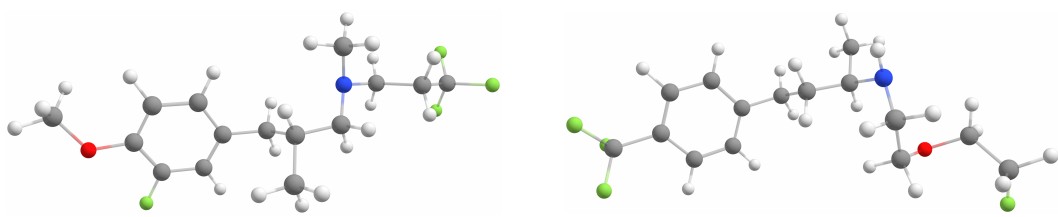

Figure 6: A pair of isomers (distinct molecules) with similarity of 0.93 in the ConforFormer–OMol embedding space but 0.29 in Uni-Mol

for ConforFormer–OMol it was above 83%, with most of the errors coming from the low capability of the model to recognize diastereomers (on backbone isomers its precision at 50% recall was 94%).

Notably, post-training the model on the train part of the PharmIsomer dataset saturates the backbone part of the benchmark with 99.9% precision at 50% recall but still reaches just 56% precision at 50% recall for diastereomers. For both isomers and diastereomers, the precision of the model is higher than of the industry standard Tanimoto similarity which relies on building a fingerprint of a molecule by matching 1024 small subgraphs against it. This representation has 100% recall by design at similarity 1, but it cannot be adjusted to obtain higher precision.

The precision and recall curves (Figure 4) for recognizing isomers of molecules outside of both Uni-Mol and OMol training datasets conclusively show that our model has obtained the capability to make inference about unique chemical structures without being directly trained on molecular graphs. While Uni-Mol replicate model seems to consider overall shape of the molecule more in making these assessments, ConforFormer–OMol recognizes the similarity based on underlying molecular graph which it inferred from a weakly supervised training. See Figure 5 for an example of conformers with very dissimilar shape and Figure 6 for a pair of isomers with an overall similar one. Both have the same similarity of 0.93 in the Uni-Mol embedding space but differ strongly (0.99 vs 0.26) in the ConforFormer–OMol one. Section F of the Supporting Information contains other examples of the models' disagreements in similarity evaluations for conformer and isomer pairs.

## 5 CONCLUSIONS AND FUTURE WORK

In this paper we obtained a compact vector representation for molecules from a weakly supervised training on molecular geometries of organic molecules. This embedding is both directly useful for molecular similarity and property prediction and shows an emergent capability to recognize molecular graph-like features from 3D geometries alone.

As directions for the future work, we want to explore molecular dynamics for automated generation of conformer/isomer labeling data for organometallic compounds; improve embeddings by incorporating additional training objectives such as the formation energy of the molecule; attempt better modeling of the conformational distributions; and research better backbone architectures and adaptors for the frozen model to improve performance.

## REPRODUCIBILITY STATEMENT

All of the code used to pre-train the models, fine-tune them, build the contrastive benchmarks and datasets, measure the results reported in Tables 1 and 2, and plot Figures 3 and 4 is available for the purposes of the double-blind anonymous review at an anonymous GitHub repository `https://github.com/ConfReview/ConforFormerReview`. The model weights are published to HuggingFace `https://huggingface.co/ConforFormer/ConforFormer`. A sample of the PharmIsomer dataset is available in the ConforFormerReview GitHub repository and will be published in full under a CC-BY license once the anonymity requirement is lifted, on a storage platform which can support its size (100+ Gb). The model training hyperparameters, dataset descriptions, and ablation study details can be found in the Supporting Information after the references.

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

## Supporting Information

## A Training details

### A.1 General Remarks

All code is available for the purposes of the double-blind anonymous review at an anonymous GitHub repository `https://github.com/ConfReview/ConforFormerReview`. The model architecture throughout this whole document is that of the standard Uni-Mol up until contrastive learning is done. This can be found in Appendix C (Table 6) in Zhou et al. (2022). Set-up is kept identical, regardless of whether the Uni-Mol, OMol, or contrastive benchmark is used as training data. The pre-trained models parameters can be found in the following HuggingFace repository `https://huggingface.co/ConforFormer/ConforFormer` The settings for the Uni-Mol replication and ConforFormer models are listed in A.2 and A.3 respectively. A three-layer $512 \times 256 \times 128$ MLP was used for fine-tuning with exactly the same settings as in Zhou et al. (2022). See the Ablation Studies (section E in the Supporting Information) for details of other experiments.

### A.2 Uni-Mol replication

Hyperparameters

- `masked_token_loss = 1`
- `masked_coord_loss = 5`
- `masked_dist_loss = 10`
- `x_norm_loss = 0.01`
- `delta_pair_repr_norm_loss = 0.01`

- `mask_prob = 0.15`
- `noise_type = "uniform"`
- `noise = 1.0`
- `only_polar = 0` (no hydrogens on the molecule)
- `dropout = 0.1` (applied to FFN, attention heads, etc.)
- Activation functions are always GeLU
- `batch size = 128`

TRAINING DETAILS

- Linear learning rate schedule
- 10,000 warm-up steps
- 1,000,000 total steps
- Validation every 10,000 steps
- Adam optimizer
- $\epsilon = 1 \times 10^{-6}$
- $\beta = (0.9, 0.99)$
- Weight decay of $1 \times 10^{-4}$

## A.3 CONFORFORMER MODELS

All hyperparameters remained the same as in A.2 except for the learning rate and the learning rate scheduling:

- Learning rate schedule altered to `ReduceLRonPlateau`
    - Patience = 3
    - $\epsilon = 0.25$
    - lr-shrink = 0.5
- 5,000 warm-up steps
- Validation was done every 5000 steps, starting after warm-up
- Peak learning rate of $5 \times 10^{-4}$
- Batch size of 128

## A.4 REDUCED UNIMOL DATASET

For quick iterations and experiments, we used a setup which could be trained to convergence overnight on a single NVIDIA H100 GPU. For that, we chose $1/8$ of the Uni-Mol dataset by simply iterating over it and selecting every 8th data point:

```
for i, datapoint in enumerate(dataset):
    if i % 8 == 0:
        new_dataset.put(datapoint)
```

For these experiments, the hyperparameters remained the same as in A.2 except for the learning rate, learning rate schedule, and batch size, unless specified otherwise.

- Learning rate schedule altered to `ReduceLRonPlateau`
    - Patience = 2
    - $\epsilon = 0.25$
    - lr-shrink = 0.5
- 5,000 warm-up steps

- Validation was done every 5000 steps, starting after warm-up

- Peak learning rate of $5 \times 10^{-4}$

- batch size = 384

## B    PharmIsomer dataset

To construct this benchmark, we used a portion of ZINC20 (Irwin et al., 2020). We took mildly reactive relatively easy to purchase molecules, which in terms of the databases marked as having reactivity up to and including "standard" and purchase up to and including "wait OK.". We removed all the overlap with Uni-Mol and OMol datasets, all unique chemical formulas were identified through their SMILES strings. We also removed all the molecules which do not have a chemical formula to avoid any potential overlap with Uni-Mol or OMol. 10 conformers of each SMILES string were generated using RDkit and optimized using the "MMFF" forcefield. Only structurally distinct conformers were used and special attention was put to ensure that no duplicate conformers were included. To ensure that only isomeric structures are assessed for similarity, simplify inference and make the metrics between different runs comparable, batches were statically constructed beforehand rather than dynamically produced at inference time. Specifically, each batch contained 128 unique molecules, which are all isomers to each other. Each isomer had exactly 2 conformers, resulting in 256 datapoints per batch. An 80/10/10 train-test-validate split was employed for the dataset so that the performance of models trained specifically on it could be evaluated; metrics in the main text are all reported on the validation part. The dataset contains 3,261,807,960 data points in 12,741,440 batches and is freely available under CC-BY license.[1]

## C    External datasets and benchmarks

### C.1    Uni-Mol

The training split of Uni-Mol, as detailed in Zhou et al. (2022), consists of 18.8M unique molecules each with 10 conformations, resulting in ca. 190M datapoints. These 10 conformations were all generated using RDKit. On average, each datapoint has 27 heavy atoms. A large portion of the dataset consists of organic molecules. The dataset contains 67 unique heavy atoms, with C, O, and N making up greater than 95%. The remaining 5% consists almost exclusively of the halogens (F, Cl, Br, and I) along with P and S. Consequently, there are only 9 heavy atom types that have a share greater than 0.01%. The validation split consists of $\approx$100K unique molecules, again with 10 conformations each. Of the 21 unique heavy atoms, the same 9 atoms have a share greater than 0.01% of the dataset. After the reduction of the dataset, taking every eighth datapoint, the relative distribution of all heavy atoms remains the same.

### C.2    OMol

The full Open Molecules (OMol-full) dataset (Levine et al., 2025) consists of various molecules which are relevant to homogenous catalysis, electrolytes, and biomolecular systems. Structures were calculated at the $\omega$B97M-V/def2-TZVPD level of theory, resulting in unquestionably higher quality data than that found in Uni-Mol. In total there are 101M unique datapoints. On average the molecules have 26 heavy atoms. C, O, and N make up 91% of all heavy atoms. OMol-full has a larger atom variety than Uni-Mol with 83 unique atom types, 59 having a share larger than 0.01% divided amongst various charge and spin states.

Reducing the dataset to that of only molecules with at least 2 heavy atoms and at least two conformations (the setup used to train ConforFormer–OMol) results in a dataset of 8.25M unique molecules, 55M total datapoints (denoted as OMol in this paper). On average, each molecule has 6-7 unique conformations. All 83 atom types remain in the dataset and now 64 have at least a 0.01% share of all heavy atoms.

---

[1]The full dataset is 100Gb+ in size and will be made available on a public platform once the anonymity requirement is lifted.

## C.3 MOLECULENET

MoleculeNet (Levine et al., 2025) is a collection of molecular benchmarks that contains tasks relevant to Physiology, Biophysics, Physical chemistry, or Quantum mechanics. Tasks are either classification or regression-based. Classification tasks are always evaluated using ROC–AUC score, while regression tasks are either evaluated using RMSE or MAE. Generally, the benchmarks contain primarily organic molecules (compositions of C, N, O), with these atoms accounting for anywhere between 70% to almost 100% of all heavy atoms (all atoms excluding H) within a benchmark. For all tasks, SMILES strings are provided alongside the targets, with only a select few (QM$x$, $x \in \{7, 8, 9\}$, the quantum mechanics based benchmarks) having provided 3D coordinates. As such, the structures generated for every 3D coordinate are those already made by the team of UniMol. In the cases where 3D coordinates are provided, only one per molecule is present, and 9 more were generated. The benchmarks are used as provided by the UniMol team.

### C.3.1 BACE

BACE is a classification benchmark with 1 target and contains slightly over 1500 molecules. The target is a binary label which qualitatively describes a molecule's ability to inhibit the human beta-secretase 1 (BACE-1). The molecules within the benchmark are purely organic, containing on average 34 heavy atoms, primarily C, N, O, and S. The halogens account for slightly over 1% of heavy atoms.

### C.3.2 BBBP

BBBP is a classification benchmark with labels indicating whether a molecule can or cannot penetrate the blood-brain barrier. The $\approx 2000$ molecules are primarily organics and occasionally halogenated. There are small amounts of salts, specifically alkai (earth) metals with 21 Na atoms and 1 Ca atom. On average, they contain 24 heavy atoms.

### C.3.3 CLINTOX

Clintox is a classification task, describing drug-like molecules using qualitative data of those approved by the FDA or failed due to toxicity. It contains $\approx 1500$ molecules, primarily of organic nature, with 26 heavy atoms. There are small amounts of main-group and d-block atoms. Curiously, all of this atomic variety is found in either the train or test splits.

### C.3.4 ESOL

ESOL is a benchmark of 1100 small ($\approx 13$ heavy atoms) organic molecules, occasionally halogenated (F, Cl, Br and I accounting for $\approx 5\%$ of heavy atoms, primarily Cl). The target is the log solubility of a molecule in water (in mol/L). It is evaluated using RMSE. Disproportionally few heavier halogens (Br and I) are in the test benchmark, specifically none.

### C.3.5 FREESOLV

FreeSolv is a benchmark of small organic molecules and their experimental or calculated solvation energy in water (in kcal/mol). Performance is evaluated using RMSE. It contains 642 small organic molecules (on average 9 heavy atoms), occasionally halogenated.

### C.3.6 HIV

HIV is a classification benchmark asking a model to distinguish between molecules which do or do not inhibit HIV replication. It is primarily organic molecules (C, O, N accounting for 95% of all heavy atoms), however still contains a large variety of alkali (earth) metals, d-block metals (frequently containing almost all occurrences of a d-block metal through the whole of MoleculeNet), and main-group elements. On average, the 41K datapoints contain approx. 25 heavy atoms.

### C.3.7 LIPO

Lipo is a regression benchmark requiring a model to predict experimentally determined log(P) (octanol/water partition coefficient) at a pH of 7.4. It 4,200 organic molecules with on average 27 heavy atoms. The performance of a model is evaluated using RMSE.

### C.3.8 MUV

MUV is a benchmark of 93K organic molecules of 24 heavy atoms on average. It contains the classification task of 17 targets, determined through high-through put experiments on BioAssays. It is a subset of datapoints contained in PCBA, refined through nearest-neighbor analysis and is meant to validated virtual screening methods.

### C.3.9 PCBA

PCBA is a benchmark of selected PubChem BioAssay consisting of results of high-throughput experiments on the biological activity of small molecules. It has 128 targets. On average, each datapoint has 26 heavy atoms, primarily C, N ,O (accounting for 95% of all heavy atoms in the benchmark), and various main-group and d-block elements. It contains 438K datapoints. The fine-tuning result of ConforFormer–OMol on this benchmark was 0.829 (on par with literature data) but due to its size we did not run it for most of the models in the study and do not include it in the tables.

### C.3.10 QM7, QM8, QM9

The QMx benchmarks consist of small organic (QM8 and QM9 occasionally halogenated) molecules with 7, 8, or 9 heavy atoms. They are regression benchmarks, performance measured in MAE, with the aim to predict various quantum-mechanically properties, such as atomization energy and HOMO-LUMO gap. They contain approx. 7K, 22K, and 130K datapoints, respectively.

### C.3.11 SIDER

SIDER is a classification benchmark with 27 targets, aiming to predict if a marketed drug has adverse drug reactions to 27 system organ classes. It consists of generally large molecules (average of 33 heavy atoms), primarily of organic nature. However, it does contain a variety of main-group and d-block elements. This atomic variety finds itself almost exclusively in the training split, frequently over 95% of these atoms. It contains 1427 datapoints.

### C.3.12 TOX21

Tox21 is a classification benchmark of 7831 datapoints and has 12 targets. These targets are binary labels as to whether a molecule has any qualitative toxicity on 12 biological systems. Datapoints contain 18.5 heavy atoms on average, primarily organic in nature. There are also small amount of main-group and d-block elements, primarily found within the training data ( $> 90\%$ occurrence in training split).

### C.3.13 TOXCAST

Toxcast has 617 binary classification targets representing qualitative toxicology data generated using in-vitro high-throughput experimentation. The ca. 8600 datapoints usually contain 19 heavy atoms, primarily C, O, N, and halogens. Main-group and d-block elements make up about 1% of all heavy atoms.

## D LLM USAGE

Large language models were used for grammar checking, LATEX formatting, initial literature search and generation of the SQLite processing code (used to support isomer classification analysis).

# E   Ablation studies

## E.1   Overview

The table S1 contains details of the specific experiments we ran, with the following table S2 containing benchmark results for various pre-training and post-training setups with different number of unfrozen layers (15 corresponding to a fully unfrozen model). The"ConforFormer" objective refers to the loss $\mathcal{L}_{\text{total}}$ as described in 3.3. Abbreviated names for experiments are used throughout these tables, with the following mapping to the main text entries:

- **U**: Uni-Mol replicate
- **U-no-flat**: Uni-Mol no "flat"
- **O-c**: Uni-Mol, OMol data
- **CF-O-c**: ConforFormer–OMol
- **CF-U**: ConforFormer–UniMol
- **Random-w**: Uni-Mol no pretrain

An important note from the ablation study is that the addition of contrastive learning in-post provides no improvement to the model. In fact, it actually worsens performance. Comparing the Cpost runs in Table S2, an increase in temperature ($\tau$) is seen to worsen finetuning results. Taking the results of BBBP as an example, the ROC–AUC score changes as $0.656 \rightarrow 0.562 \rightarrow 0.544$ as $\tau$ goes from 0.01 to 0.1 and finally 0.5. We used these results to guide our choice of $\tau$ for ConforFormer, but we checked that increasing the temperature of the NT-Xent loss in pre-training worsens the performacne as well (see **CF-U-r-0.25**). As the model is made to differentiate more, it performs worse. Compared to the results of "ConforFormer–OMol", the best performing model with contrastive pose-training results in worse metrics after finetuning.

A contrastive-only pre-training on the Unimol dataset (**Contrast-U-r**), on the other hand, results in a below average quality of the embeddings but still performs surprisingly well, suggesting that contrastive objective alone could also be potentially viable strategy for model pre-training.

Sanity checks were run to validate that the ConforFormer loss and not changes in the training setup were actually driving the metric improvements. However, changing the batch size (**U-r-128**) or adding more conformers in each batch (**U-r-256-conf**) did not lead to any noticeable improvements. Training the Uni-Mol model without additional contrastive loss on the full OpenMolecules dataset (**O**) did not improve benchmarks beyond ConforFormer–OMol **CF-O-c** or Uni-Mol replicate **U** either.

Given the results of models trained using contrastive learning on PharmIsomer, it was hypothesized that doing training that mimics this benchmark would improve performance. For this, the Uni-Mol data set was tailored to allow for the construction of batches that guarantee a minimum number of isomers. The results of these pre-training can be found in the **CF-U-$x$I** setups, $x \in \{10, 25, 40\}$ in Table S2 . These 3 rows represent a minimum of 40%, 25%, and 10% of datapoints within a batch having an isomer pair. No noticeable improvement is observed within the classification tasks and no significant trend is observed for the regression tasks. Thus, the hypothesis of additional isomers in the batch is not seen to be true for the Uni-Mol dataset with the contrastive training set-up outlined in Section A.3.

Each line in Table S2 corresponds to a single fine-tuning run. Please refer to the subsection 3.2 of the main text for the stability analysis.

## E.2   Full run data

The labels for pre-training runs from Table S1 are used throughout Table S2, which contains fine-tuning results with different number of model layers unfrozen. Layers were frozen starting from the last. Finetuning was performed using the hyperparameters specified in the Uni-Mol GitHub repository https://github.com/deepmodeling/Uni-Mol/tree/main/unimol. A typical fine-tuning job took $\approx 3$ hours on an A100 GPU. For the fine-tuning starting from **Random-w**, all batch sizes were set to 128, and training was stopped after no improvement was made after 40

epochs (18 hours on an H100). The remaining hyperparameters are identical to those under standard finetunings.

Table S1: Training setups used in the experiments. Batch size refers to the pre-training procedure; when written as $n_u$ $(n_t)$, it denotes $n_t$ geometries total in the batch, with $n_u$ used to compute the Uni-Mol losses $\mathcal{L}_{\text{token}}$, $\mathcal{L}_{\text{coord}}$, and $\mathcal{L}_{\text{distance}}$. PharmIsomer post-training batch size, where applicable, is always 256.

| Training | Dataset | GPU | Hours | Objective | Batch size |
|---|---|---|---|---|---|
| CF-2U-r | 1/8 Uni-Mol | H100 | 14 | ConforFormer + Uni-Mol loss on full batch, $\tau = 0.07$ | 256 |
| CF-O-c | OMol | 4×H100 | 24 | ConforFormer, $\tau = 0.07$ | 128 (256) |
| CF-U | Uni-Mol | H100 | 48 | ConforFormer, $\tau = 0.07$ | 128 (256) |
| CF-U-10I | Uni-Mol, 10% isomers | H100 | 48 | ConforFormer, $\tau = 0.07$ | 128 (256) |
| CF-U-25I | Uni-Mol, 25% isomers | H100 | 48 | ConforFormer, $\tau = 0.07$ | 128 (256) |
| CF-U-40I | Uni-Mol, 40% isomers | H100 | 48 | ConforFormer, $\tau = 0.07$ | 128 (256) |
| CF-U-H | Uni-Mol with H | A100 | 72 | ConforFormer, $\tau = 0.07$ | 128 (256) |
| CF-U-r | 1/8 Uni-Mol | A100 | 14 | ConforFormer, $\tau = 0.07$ | 128 (256) |
| CF-U-r-0.25 | 1/8 Uni-Mol | A100 | 14 | ConforFormer, $\tau = 0.25$ | 128 (256) |
| Contrast-U-r | 1/8 Uni-Mol | A100 | 10 | Contrast loss only, $\tau = 0.07$ | 256 |
| O | OMol-full | A100 | 120 | Uni-Mol | 384 |
| O-c | OMol | A100 | 120 | Uni-Mol | 128 |
| O-Cpost-0.01 | OMol-full | A100 | 120 | Uni-Mol + contrast on PharmIsomer in post-train, $\tau = 0.01$ | 384 |
| O-Cpost-0.05 | OMol-full | A100 | 120 | Uni-Mol + contrast on PharmIsomer in post-train, $\tau = 0.05$ | 384 |
| Random-w | – | A100 | – | Random weight initialization | – |
| U | Uni-Mol | A100 | 72 | Uni-Mol | 128 |
| U-no-flat | Uni-Mol, no "flat" | A100 | 72 | Uni-Mol | 128 |
| U-r | 1/8 Uni-Mol | A100 | 10 | Uni-Mol | 384 |
| U-r-128 | 1/8 Uni-Mol | A100 | 14 | Uni-Mol | 128 |
| U-r-256 | 1/8 Uni-Mol | H100 | 14 | Uni-Mol | 256 |
| U-r-256-conf | 1/8 Uni-Mol, 2 conformers each | H100 | 14 | Uni-Mol | 256 |
| U-r-Cpost-0.01 | 1/8 Uni-Mol | H100 | 14 | Uni-Mol + contrast on PharmIsomer in post-train, $\tau = 0.01$ | 384 |
| U-r-Cpost-0.1 | 1/8 Uni-Mol | H100 | 14 | Uni-Mol + contrast on PharmIsomer in post-train, $\tau = 0.1$ | 384 |
| U-r-Cpost-0.5 | 1/8 Uni-Mol | H100 | 14 | Uni-Mol + contrast on PharmIsomer in post-train, $\tau = 0.5$ | 384 |

Table S2: Benchmark results on MoleculeNet as detailed in C.3. "Unfrozen" refers to the number of model layers unfrozen diring the fine-tuning procedure. Left block: classification benchmarks. Right block: regression benchmarks.

| Training | Unfrozen | ROC-AUC, ↑ | | | | | | | | RMSE, ↓ | | | | | |
|---|---|---|---|---|---|---|---|---|---|---|---|---|---|---|---|
| | | BBBP | BACE | ClinTox | Tox21 | ToxCast | SIDER | HIV | MUV | ESol | FreeSolv | Lipo | QM7 | QM8 | QM9 |
| CF-2U-r | 0 | 0.671 | 0.710 | 0.589 | 0.715 | 0.603 | 0.646 | 0.678 | 0.771 | 1.24 | 3.52 | 0.94 | 88.2 | 0.0236 | 0.0143 |
| CF-O-c | 0 | 0.673 | 0.763 | 0.724 | 0.753 | 0.638 | 0.643 | 0.724 | 0.739 | 1.04 | 3.67 | 0.75 | 163.0 | 0.0223 | 0.0123 |
| CF-U | 0 | 0.626 | 0.785 | 0.596 | 0.736 | 0.617 | 0.623 | 0.603 | 0.747 | 1.06 | 3.66 | 0.87 | 93.4 | 0.0243 | 0.0140 |
| CF-U-10I | 0 | 0.657 | 0.754 | 0.586 | 0.732 | 0.620 | 0.603 | 0.738 | 0.744 | 1.34 | 3.39 | 0.91 | 105.4 | 0.0262 | 0.0163 |
| CF-U-25I | 0 | 0.646 | 0.752 | 0.651 | 0.709 | 0.612 | 0.593 | 0.702 | 0.746 | 1.48 | 3.87 | 0.88 | 98.7 | 0.0250 | 0.0152 |
| CF-U-40I | 0 | 0.660 | 0.732 | 0.518 | 0.694 | 0.606 | 0.596 | 0.685 | 0.780 | 1.51 | 4.24 | 0.91 | 103.5 | 0.0254 | 0.0170 |
| CF-U-H | 0 | 0.641 | 0.799 | 0.430 | 0.721 | 0.626 | 0.638 | 0.707 | 0.709 | 1.18 | 3.60 | 0.84 | 119.3 | 0.0248 | 0.0140 |
| CF-U-r | 0 | 0.658 | 0.760 | 0.530 | 0.752 | 0.645 | 0.650 | 0.707 | 0.714 | 1.13 | 3.36 | 0.80 | 97.3 | 0.0227 | 0.0129 |
| CF-U-r-0.25 | 0 | 0.583 | 0.657 | 0.685 | 0.674 | 0.564 | 0.608 | 0.722 | 0.644 | 1.57 | 3.93 | 1.00 | 97.3 | 0.0296 | 0.0242 |
| Contrast-U-r | 0 | 0.635 | 0.751 | 0.566 | 0.688 | 0.590 | 0.589 | 0.695 | 0.684 | 1.35 | 4.02 | 0.98 | 117.2 | 0.0259 | 0.0171 |
| O | 0 | 0.655 | 0.775 | 0.630 | 0.693 | 0.639 | 0.583 | 0.723 | 0.742 | 1.17 | 2.95 | 1.03 | 109.5 | 0.0309 | 0.0257 |
| O-c | 0 | 0.659 | 0.787 | 0.680 | 0.710 | 0.629 | 0.613 | 0.758 | 0.695 | 1.18 | 3.01 | 0.95 | 85.6 | 0.0278 | 0.0200 |
| O-Cpost-0.01 | 0 | 0.672 | 0.722 | 0.562 | 0.679 | | 0.609 | 0.666 | 0.573 | 1.52 | 4.29 | 1.03 | 91.4 | 0.0270 | 0.0191 |
| O-Cpost-0.05 | 0 | 0.613 | 0.783 | 0.768 | 0.682 | 0.574 | 0.548 | 0.576 | 0.603 | 1.71 | 4.30 | 1.06 | 96.9 | 0.0280 | 0.0221 |
| U | 0 | 0.633 | 0.778 | 0.753 | 0.717 | 0.652 | 0.583 | 0.732 | 0.738 | 1.16 | 2.59 | 0.92 | 89.3 | 0.0274 | 0.0208 |
| U-no-flat | 0 | 0.650 | 0.778 | 0.727 | 0.712 | 0.636 | 0.598 | 0.744 | 0.734 | 1.27 | 2.89 | 0.93 | 91.4 | 0.0264 | 0.0191 |
| U-r | 0 | | 0.728 | 0.742 | 0.671 | 0.626 | | 0.757 | 0.603 | 1.19 | 2.86 | 1.03 | 113.9 | 0.0297 | 0.0241 |
| U-r-128 | 0 | 0.673 | 0.777 | 0.746 | 0.655 | | 0.582 | 0.743 | 0.534 | 1.51 | 3.19 | 1.04 | 127.1 | 0.0315 | 0.0266 |
| U-r-256 | 0 | 0.654 | 0.771 | 0.626 | 0.677 | 0.612 | 0.602 | 0.734 | 0.641 | 1.29 | 2.75 | 1.02 | 143.8 | 0.0297 | 0.0255 |
| U-r-256-conf | 0 | 0.654 | 0.764 | 0.626 | 0.677 | 0.612 | 0.607 | 0.734 | 0.641 | 1.33 | 2.75 | 1.04 | 116.1 | 0.0287 | 0.0232 |
| U-r-Cpost-0.01 | 0 | 0.656 | 0.730 | 0.438 | 0.692 | 0.601 | | 0.660 | 0.626 | 1.24 | 3.37 | 0.99 | 81.1 | 0.0273 | 0.0207 |
| U-r-Cpost-0.1 | 0 | 0.562 | 0.679 | 0.574 | 0.591 | 0.522 | | 0.559 | 0.539 | 1.89 | 4.00 | 1.12 | 126.5 | 0.0318 | 0.0274 |
| U-r-Cpost-0.5 | 0 | 0.544 | 0.590 | 0.500 | 0.603 | 0.540 | 0.540 | 0.518 | 0.640 | 1.99 | 4.39 | 1.11 | 126.4 | 0.0320 | 0.0283 |
| CF-O-c | 1 | 0.676 | 0.811 | 0.682 | 0.774 | 0.667 | 0.655 | 0.773 | 0.824 | 1.08 | 2.38 | 0.67 | 96.0 | 0.0188 | 0.0083 |
| CF-U | 1 | 0.659 | 0.802 | 0.650 | 0.773 | 0.662 | 0.638 | 0.763 | 0.793 | 0.95 | 2.92 | 0.75 | 101.9 | 0.0216 | 0.0100 |

*Continued on next page*

| Training | Unfrozen | ROC-AUC, ↑ | | | | | | | | RMSE, ↓ | | | | | |
|---|---|---|---|---|---|---|---|---|---|---|---|---|---|---|---|
| | | BBBP | BACE | ClinTox | Tox21 | ToxCast | SIDER | HIV | MUV | ESol | FreeSolv | Lipo | QM7 | QM8 | QM9 |
| CF-U-H | 1 | 0.641 | 0.828 | 0.727 | 0.751 | 0.663 | 0.652 | 0.766 | 0.805 | 1.00 | 2.62 | 0.73 | 98.9 | 0.0217 | 0.0101 |
| CF-U-r | 1 | 0.683 | 0.805 | 0.656 | 0.775 | 0.682 | 0.590 | 0.757 | 0.775 | 1.05 | 2.16 | 0.74 | 88.1 | 0.0187 | 0.0082 |
| CF-U-r-0.25 | 1 | 0.643 | 0.673 | 0.641 | 0.718 | 0.624 | 0.624 | 0.766 | 0.698 | 1.31 | 2.63 | 0.85 | 78.9 | 0.0188 | 0.0091 |
| O | 1 | 0.651 | 0.799 | 0.596 | 0.751 | 0.654 | 0.671 | 0.736 | 0.730 | 1.10 | 2.64 | 0.80 | 105.9 | 0.0232 | 0.0131 |
| O-Cpost-0.01 | 1 | 0.700 | | 0.672 | 0.685 | 0.618 | 0.615 | 0.668 | 0.609 | 1.34 | 3.94 | 0.97 | 79.8 | 0.0251 | 0.0137 |
| O-Cpost-0.05 | 1 | 0.614 | 0.831 | 0.722 | 0.722 | 0.598 | 0.559 | 0.598 | 0.622 | | 4.24 | 1.02 | 91.8 | 0.0251 | 0.0145 |
| U | 1 | 0.727 | 0.779 | 0.826 | 0.780 | 0.695 | 0.632 | 0.780 | 0.813 | 0.83 | 1.88 | 0.65 | 85.5 | 0.0181 | 0.0078 |
| U-no-flat | 1 | 0.723 | 0.788 | 0.889 | 0.789 | 0.683 | 0.635 | 0.790 | 0.787 | 0.86 | 2.12 | 0.65 | 62.8 | 0.0174 | 0.0072 |
| U-r | 1 | | 0.747 | 0.708 | 0.755 | 0.665 | | 0.766 | 0.704 | 1.00 | 2.90 | 0.77 | 113.3 | 0.0219 | 0.0118 |
| U-r-Cpost-0.01 | 1 | 0.674 | 0.787 | 0.543 | 0.735 | 0.630 | 0.614 | 0.667 | | 1.06 | 2.84 | | 77.3 | 0.0235 | 0.0119 |
| U-r-Cpost-0.1 | 1 | 0.650 | 0.786 | 0.584 | 0.650 | 0.594 | 0.577 | 0.568 | 0.497 | 1.20 | 3.55 | | 85.8 | 0.0266 | 0.0163 |
| U-r-Cpost-0.5 | 1 | 0.650 | 0.786 | 0.584 | 0.710 | 0.602 | 0.596 | 0.568 | 0.497 | 1.20 | 3.55 | | 96.0 | 0.0254 | 0.0148 |
| CF-O-c | 3 | 0.702 | 0.816 | 0.670 | 0.779 | 0.668 | 0.654 | 0.784 | 0.791 | 1.02 | 2.21 | 0.67 | 72.7 | 0.0180 | 0.0067 |
| CF-U | 3 | 0.665 | 0.803 | 0.671 | 0.777 | 0.668 | 0.641 | 0.766 | 0.795 | 0.94 | 2.59 | 0.74 | 101.2 | 0.0192 | 0.0079 |
| CF-U-H | 3 | 0.639 | 0.829 | 0.723 | 0.760 | 0.672 | 0.647 | 0.762 | 0.790 | 0.96 | 2.40 | 0.70 | 90.5 | 0.0191 | 0.0079 |
| CF-U-r | 3 | 0.689 | 0.821 | 0.646 | 0.777 | 0.684 | 0.630 | 0.769 | 0.760 | 0.95 | 2.25 | 0.72 | 78.7 | 0.0177 | 0.0069 |
| CF-U-r-0.25 | 3 | 0.680 | 0.740 | 0.643 | 0.724 | 0.646 | 0.647 | 0.762 | 0.675 | 1.13 | 2.20 | 0.77 | 69.7 | 0.0176 | 0.0068 |
| O | 3 | 0.728 | 0.837 | 0.630 | 0.751 | 0.654 | 0.671 | 0.745 | 0.797 | 1.11 | 2.25 | 0.71 | 82.4 | 0.0200 | 0.0087 |
| O-Cpost-0.01 | 3 | | 0.752 | 0.681 | 0.713 | 0.622 | 0.604 | 0.687 | 0.663 | 1.20 | 3.79 | 0.93 | 64.4 | 0.0225 | 0.0098 |
| O-Cpost-0.05 | 3 | 0.653 | 0.825 | 0.775 | 0.720 | 0.609 | 0.566 | 0.664 | 0.679 | 1.23 | 4.13 | 0.97 | 68.5 | 0.0263 | 0.0102 |
| U | 3 | 0.717 | 0.785 | 0.842 | 0.782 | 0.692 | 0.636 | 0.791 | 0.807 | 0.88 | 1.63 | 0.61 | 74.3 | 0.0165 | 0.0060 |
| U-no-flat | 3 | 0.719 | 0.791 | 0.902 | 0.785 | 0.690 | 0.633 | 0.794 | 0.798 | 0.91 | 1.84 | 0.62 | 56.7 | 0.0168 | 0.0057 |
| U-r | 3 | 0.698 | 0.832 | 0.749 | 0.779 | 0.676 | 0.638 | 0.778 | 0.795 | 0.88 | 2.30 | 0.69 | 90.6 | 0.0185 | 0.0080 |
| U-r-Cpost-0.01 | 3 | 0.696 | 0.785 | 0.579 | | 0.629 | 0.624 | 0.693 | | 0.98 | 2.42 | | 80.8 | 0.0205 | 0.0088 |
| U-r-Cpost-0.1 | 3 | 0.671 | 0.785 | | 0.683 | 0.608 | 0.584 | 0.613 | 0.582 | 1.10 | 3.09 | 1.02 | 78.0 | 0.0234 | 0.0105 |
| U-r-Cpost-0.5 | 3 | 0.661 | 0.661 | 0.503 | 0.729 | 0.605 | 0.595 | 0.578 | 0.602 | 1.08 | 3.74 | 0.93 | 93.6 | 0.0231 | 0.0100 |
| CF-O-c | 15 | 0.650 | 0.807 | 0.716 | 0.778 | 0.691 | 0.649 | 0.782 | 0.772 | 0.94 | 2.15 | 0.64 | 60.3 | 0.0159 | 0.0054 |
| CF-U-r | 15 | 0.699 | 0.815 | 0.745 | | | | 0.789 | 0.819 | 0.88 | 1.89 | 0.66 | 59.4 | 0.0168 | |

*Continued on next page*

| Training | Unfrozen | ROC-AUC, ↑ | | | | | | | | RMSE, ↓ | | | | | |
| --- | --- | --- | --- | --- | --- | --- | --- | --- | --- | --- | --- | --- | --- | --- | --- |
| | | BBBP | BACE | ClinTox | Tox21 | ToxCast | SIDER | HIV | MUV | ESol | FreeSolv | Lipo | QM7 | QM8 | QM9 |
| O | 15 | 0.680 | 0.859 | 0.704 | 0.785 | 0.680 | 0.618 | 0.785 | 0.711 | 0.87 | 1.95 | 0.66 | 52.0 | 0.0159 | 0.0054 |
| Random-w | 15 | 0.639 | 0.823 | 0.592 | 0.746 | 0.648 | 0.617 | 0.761 | 0.619 | 0.98 | 2.83 | 0.77 | 97.3 | 0.0178 | 0.0063 |
| U | 15 | 0.723 | 0.811 | 0.861 | 0.786 | 0.684 | 0.646 | 0.771 | 0.789 | 0.81 | 1.93 | 0.62 | 59.2 | 0.0161 | 0.0052 |
| U-no-flat | 15 | 0.694 | 0.796 | 0.883 | 0.797 | 0.688 | 0.648 | 0.800 | 0.883 | 0.74 | 3.08 | 0.38 | 64.2 | 0.0160 | 0.0053 |
| U-r | 15 | 0.725 | 0.825 | 0.854 | 0.782 | 0.677 | 0.637 | 0.780 | 0.792 | 0.85 | 2.15 | 0.64 | 58.1 | 0.0166 | 0.0060 |

# F    MORE VISUAL EXAMPLES OF MODEL PERFORMANCE

## F.1    ISOMERS

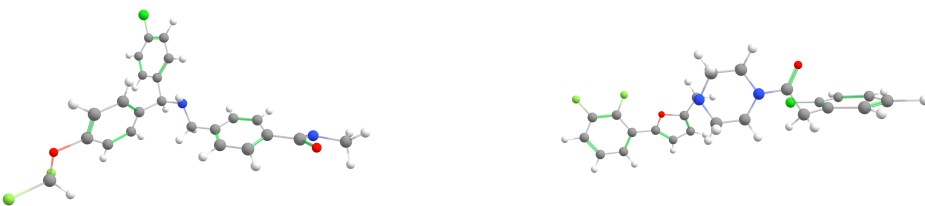

Figure S1: A pair of isomers (distinct molecules) having similarity of 0.99 in the Uni-Mol embedding space and 0.04 in ConforFormer–OMol

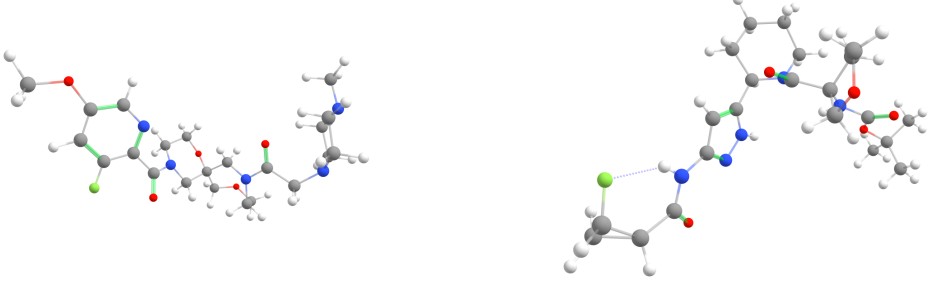

Figure S2: A pair of isomers (distinct molecules) having similarity of 0.99 in the Uni-Mol embedding space and 0.20 in ConforFormer–OMol

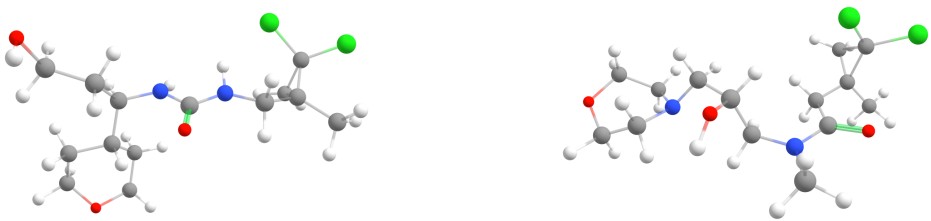

Figure S3: A pair of isomers (distinct molecules) having similarity of 0.99 in the Uni-Mol embedding space and 0.70 in ConforFormer–OMol

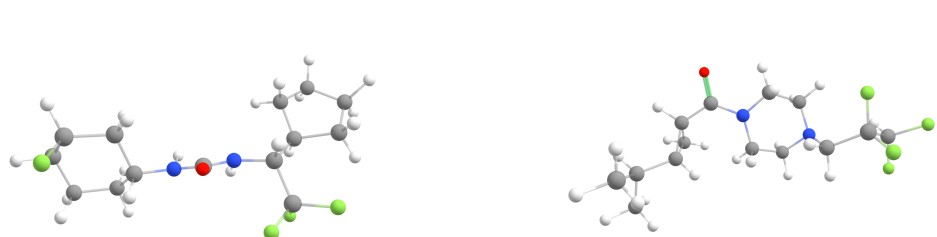

Figure S4: A pair of isomers (distinct molecules) having similarity of 0.98 in the Uni-Mol embedding space and 0.40 in ConforFormer–OMol

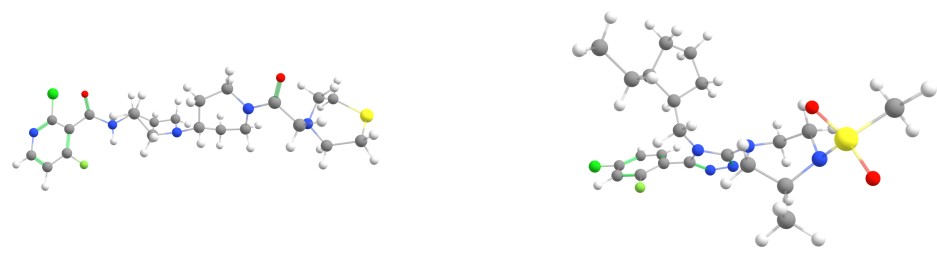

Figure S5: A pair of isomers (distinct molecules) having similarity of 0.98 in the Uni-Mol embedding space and 0.20 in ConforFormer–OMol

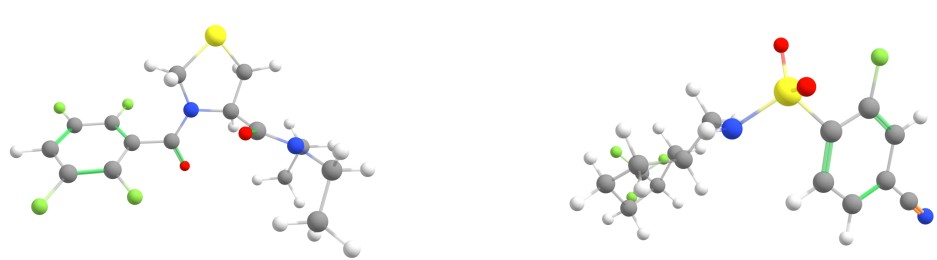

Figure S6: A pair of isomers (distinct molecules) having similarity of 0.93 in the Uni-Mol embedding space and 0.14 in ConforFormer–OMol

## F.2 CONFORMERS

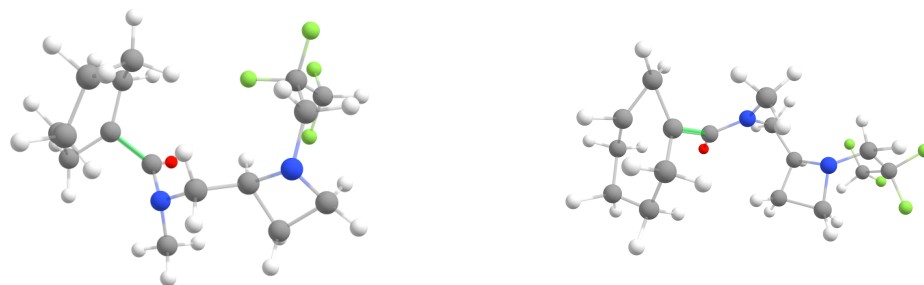

Figure S7: A pair of conformers of the same molecule having similarity of 0.95 (very low, below the 2nd percentile of all pairs) in the Uni-Mol embedding space and 0.99 in ConforFormer–OMol

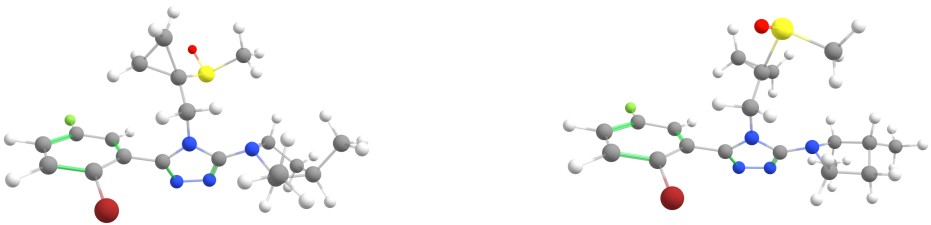

Figure S8: A pair of conformers of the same molecule having similarity of 0.95 (very low, below the 2d percentile ofd all pairs) in the Uni-Mol embedding space and 0.99 in ConforFormer–OMol

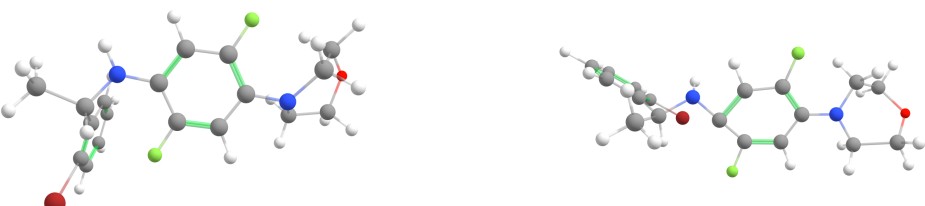

Figure S9: A pair of conformers of the same molecule having similarity of 0.96 in the Uni-Mol embedding space and 0.64 in ConforFormer–OMol. *Note* the distorted ring in the right image is an indication of improperly generated data and is technically not a conformer of the image left. This chemically significant distortion in the structure is not detected by Uni-Mol.

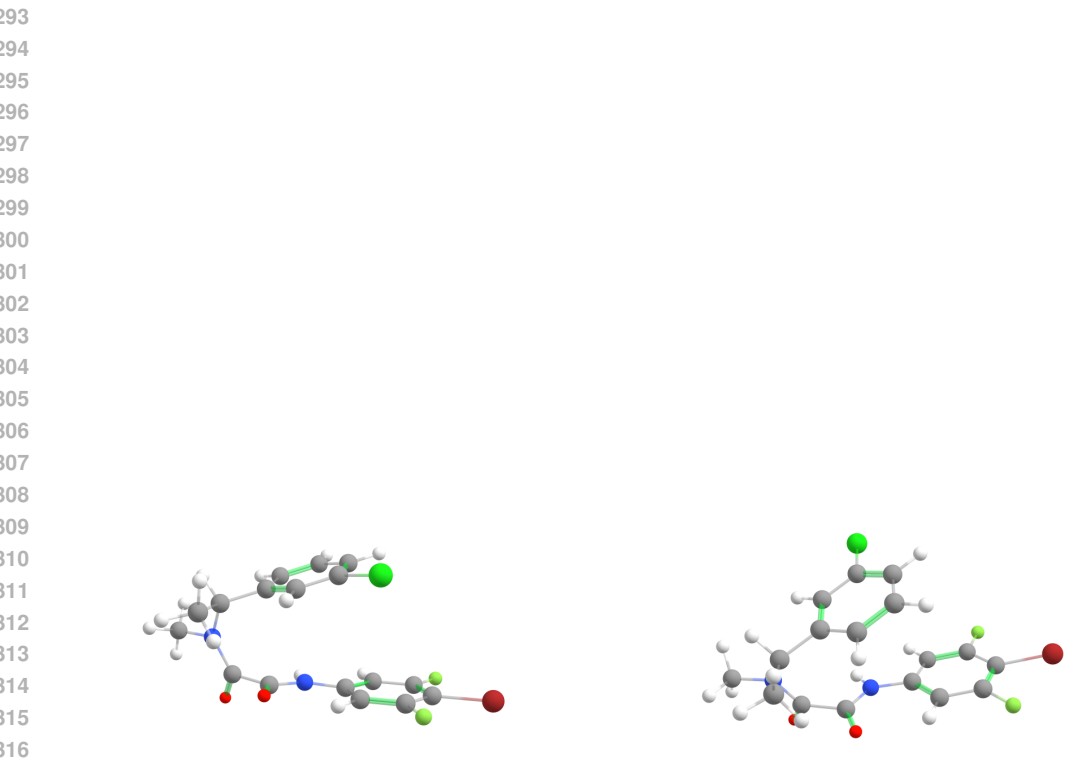

Figure S10: A pair of conformers of the same molecule having similarity of 0.94 in the Uni-Mol embedding space and 0.99 in ConforFormer–OMol.

