# OpenReview forum: "ConforFormer: Representation for Molecules Through Understanding of Conformers"
_ICLR.cc/2026/Conference — Submitted to ICLR 2026_

### Official Review · Reviewer_a89c · 2025-10-29

**Soundness:** 2
**Presentation:** 2
**Contribution:** 1
**Rating:** 2
**Confidence:** 4

**Summary:**

The paper modifies Uni-Mol by retraining the model without “flat” structures (molecules with all Z coordinates set to zero), finding minimal impact on performance; introducing a contrastive learning objective to align embeddings across multiple 3D conformations, though the novelty over Uni-Mol’s existing contrastive loss is unclear; and exploring the use of the Organic Molecules (OMol) dataset instead of the original Uni-Mol dataset, which did not lead to significant improvements. Overall, the method consists of incremental adjustments rather than a fundamentally new approach.

**Strengths:**

The paper is detailed and uses domain-specific terminology appropriately.

**Weaknesses:**

Overall, the paper reads more like a technical report on modifications to Uni-Mol rather than presenting a strong methodological contribution.

- **Modification of Uni-Mol:**
   - The authors identified that the original Uni-Mol model included “flat” structures (atoms with Z = 0) in ~9% of the training data and retrained the model without them, finding negligible performance degradation.
While this is a useful insight, it is minor. The original design helps Uni-Mol generalize to molecules with incomplete 3D information or planar molecules (e.g., aromatic rings, graphene fragments). Showing that removing this feature has little effect is interesting but not a major contribution.
  - The paper proposes a contrastive learning objective to align embeddings across multiple conformations. However, Uni-Mol already applies contrastive learning across conformations, where positive pairs are conformations of the same molecule and negatives otherwise.
It is unclear how the proposed contrastive loss differs from the original Uni-Mol approach, and this should be clarified.
  - The exploration of the Organic Molecules (OMol) dataset instead of the original Uni-Mol dataset did not lead to noticeable performance improvements.
- The authors do not include a baseline using only graph data (without 3D information). Without this, the effectiveness of 3D conformations cannot be properly evaluated, especially when some works has shown that using RDKit-generated 3D coordinates can even lead to degrade performance (RDKit is not very accurate in this task).
- **Terminology and ML Understanding:**
  - The manuscript misuses some CS/ML terminology. For example, “retrain” is incorrectly used where “fine-tune” is meant (e.g., abstract: *current approaches require retraining the entire model for each prediction task, using published weights only as initialization* → this is actually fine-tuning).
  - Moreover, the statement *current approaches require retraining the entire model for each prediction task, using published weights only as initialization* is inaccurate at a deeper level: pretrained models can be used as feature extractors to train other models for downstream tasks. This is one of the main motivations of the paper, undermining the importance of the actual problem they are trying to solve. Overall, this suggests a weak understanding of ML concepts.
  - The claim *from a physical point of view, molecular graphs do not exist* is unconvincing as an argument against this datatype. Many representations (FASTA sequences, DNA sequences, or even text) do not physically exist in the same sense, yet are useful abstractions. This argument does not support the proposed approach.
  - The claim *structural formulas work well for the chemistry of organic molecules, but for more complex compounds* is misleading. The sentence implies that *more complex compounds* (such as organometallics) are not organic molecules, which is not entirely accurate. Moreover, I think this work focuses on building models for representing organic compounds.
  - The term *task-agnostic* in the abstract is an overstatement, since the authors still fine-tune their models on downstream tasks.

**Questions:**

How does your use of contrastive learning differ from that in Uni-Mol?

---

> ### Author Response · Authors · 2025-11-24
> **Reply to Reviewer a89c (part 1)**
>
> Thank you for the review! Please find the replies to your concerns below.
>
> > •	Modification of Uni-Mol:
>
> >o	The authors identified that the original Uni-Mol model included “flat” structures (atoms with Z = 0) in ~9% of the training data and retrained the model without them, finding negligible performance degradation. While this is a useful insight, it is minor. The original design helps Uni-Mol generalize to molecules with incomplete 3D information or planar molecules (e.g., aromatic rings, graphene fragments). Showing that removing this feature has little effect is interesting but not a major contribution.
>
> >o	The paper proposes a contrastive learning objective to align embeddings across multiple conformations. However, Uni-Mol already applies contrastive learning across conformations, where positive pairs are conformations of the same molecule and negatives otherwise. It is unclear how the proposed contrastive loss differs from the original Uni-Mol approach, and this should be clarified.
>
> >o	The exploration of the Organic Molecules (OMol) dataset instead of the original Uni-Mol dataset did not lead to noticeable performance improvements.
>
> 1)	Thank you. Given that we focused on working with 3D geometries produced by molecular modeling, “molecules with incomplete 3D information” were outside the scope of our research.
> 2)	We kindly disagree with the reviewer regarding the statement “Uni-Mol already applies contrastive learning across conformations”. Neither the main paper text nor the Uni-Mol code has any mention of this; contrastive learning is listed as an unsuccessful experiment in the Supporting information.
> 3)	The dataset is called OpenMolecules, and training on it did in fact improve the quality of the frozen embeddings, as described in Tables 1 and 2 in the paper text.
>
> > •	The authors do not include a baseline using only graph data (without 3D information). Without this, the effectiveness of 3D conformations cannot be properly evaluated, especially when some works has shown that using RDKit-generated 3D coordinates can even lead to degrade performance (RDKit is not very accurate in this task).
>
> Thank you for the suggestion, we will explore this aspect in future work. However, the focus of this paper was to recover molecular-graph-equivalent representation from 3D coordinates without explicitly supplying the molecular graph info. Building embeddings for molecular graphs is an approach explored in many papers, starting with N-Gram Graph (S Liu et al., NeurIPS  2019).
>
> *(continued in part 2 due to the character limit)*

---

> > ### Author Response · Authors · 2025-11-24
> > **Reply to Reviewer a89c (part 2)**
> >
> > > •	Terminology and ML Understanding:
> >
> > > o	The manuscript misuses some CS/ML terminology. For example, “retrain” is incorrectly used where “fine-tune” is meant (e.g., abstract: current approaches require retraining the entire model for each prediction task, using published weights only as initialization → this is actually fine-tuning).
> >
> > > o	Moreover, the statement current approaches require retraining the entire model for each prediction task, using published weights only as initialization is inaccurate at a deeper level: pretrained models can be used as feature extractors to train other models for downstream tasks. This is one of the main motivations of the paper, undermining the importance of the actual problem they are trying to solve. Overall, this suggests a weak understanding of ML concepts.
> >
> > > o	The claim from a physical point of view, molecular graphs do not exist is unconvincing as an argument against this datatype. Many representations (FASTA sequences, DNA sequences, or even text) do not physically exist in the same sense, yet are useful abstractions. This argument does not support the proposed approach.
> >
> > > o	The claim structural formulas work well for the chemistry of organic molecules, but for more complex compounds is misleading. The sentence implies that more complex compounds (such as organometallics) are not organic molecules, which is not entirely accurate. Moreover, I think this work focuses on building models for representing organic compounds.
> >
> > > o	The term task-agnostic in the abstract is an overstatement, since the authors still fine-tune their models on downstream tasks.
> >
> > 1)	Thank you for the comment. We wanted to distinguish between full-model fine-tuning and few-layer finetuning but the result was probably confusing.  We corrected the usage in the abstract as follows:
> >
> > *However, current approaches require **updating the weights of the entire model during the fine-tuning procedure** for each prediction task. While this enables state-of-the-art performance, it limits practical deployment, as real-world datasets are often too small to support the stable retraining of large models.*
> >
> > The main text exclusively uses the term “fine-tuning” throughout. If you have other examples of what you consider “misuse of CS/ML terminology”, we will be happy to correct them as well.
> >
> > 2)	Our paper focuses specifically on pre-training the model to produce an embedding (feature vector) which can be re-used between different downstream tasks. This is the distinguishing feature of our work compared to previously published approaches to molecular foundational models.
> > 3)	With all respect, organometallic compounds are not organic molecules.  Moreover, there is no standard for SMILES or InCHI to handle them, suggesting that there is no common established  way to encode their structural formulas.
> > 4)	This is more of an argument for treating the 3D molecular geometries as the “primary form” of chemical data. A good analogy can be found in the domain of computer vision, which works with bitmaps and text labels. While labels (2D molecular graphs in chemical domain) are useful, real progress comes from understanding the raw data (3D geometries in the case of chemistry).
> > 5)	No, we don’t fine-tune the full model, we only train a shallow MLP on top of frozen embeddings.
> >
> > > How does your use of contrastive learning differ from that in Uni-Mol?
> >
> > As answered above, Uni-Mol does not employ contrastive learning.

---

### Official Review · Reviewer_mwTZ · 2025-10-29

**Soundness:** 1
**Presentation:** 2
**Contribution:** 1
**Rating:** 2
**Confidence:** 3

**Summary:**

This paper proposes a pretraining strategy that incorporates conformational information to enhance molecular representation learning. The authors argue that modeling 3D space is essential for capturing molecular properties. While the motivation is sound, the manuscript suffers from several critical issues that undermine its contribution.

**Strengths:**

The topic of incorporating 3D information is well-motivated.

**Weaknesses:**

1. **Misrepresentation of Prior Work**
   The authors claim that “no chemical embedding model capturing the diversity of 3D molecular conformations has yet been published.” This statement overlooks a substantial body of literature on conformer-aware pretraining. Several existing models explicitly incorporate 3D conformational diversity, and the lack of engagement with these works raises concerns about the novelty and scholarly rigor of the paper.

2. **Limited Novelty**
   The proposed techniques—pretraining on conformers and freezing the backbone while fine-tuning only the final MLP layer—are well-established practices in molecular machine learning. The manuscript does not present sufficient innovation beyond these standard approaches.

3. **Underwhelming Performance**
   As shown in Tables 1 and 2, the model's performance falls short of state-of-the-art methods across multiple benchmarks. The results do not convincingly demonstrate that the proposed approach yields meaningful improvements in molecular representation quality.

**Questions:**

**Questionable Embedding Behavior**
   Figure 5 presents a pair of conformers with substantial geometric differences. If ConforFormer-OMol had truly learned a robust understanding of 3D molecular structure, the cosine similarity between these embeddings should be significantly lower. This example casts doubt on the model’s ability to distinguish conformational nuances.

---

> ### Author Response · Authors · 2025-11-24
> **Reply to Reviewer mwTZ**
>
> Thank you for the review! Please find the replies to your concerns below.
>
> > 1.	Misrepresentation of Prior Work
> The authors claim that “no chemical embedding model capturing the diversity of 3D molecular conformations has yet been published.” This statement overlooks a substantial body of literature on conformer-aware pretraining. Several existing models explicitly incorporate 3D conformational diversity, and the lack of engagement with these works raises concerns about the novelty and scholarly rigor of the paper.
>
> The focus of the paper is specifically on producing useful molecular embeddings from 3D geometries. While some training procedures and models utilize 3D molecular geometry, there are no published embedding models aware of those (fingerprint-based and molecular-graph-based solutions do exist, mostly constructed for substructure search). Here, we refer to embeddings as frozen representations of the molecule which are then reused in diverse downstream tasks.
>
> >2.	Limited Novelty
> The proposed techniques—pretraining on conformers and freezing the backbone while fine-tuning only the final MLP layer—are well-established practices in molecular machine learning. The manuscript does not present sufficient innovation beyond these standard approaches.
>
> We are not aware of the literature that the referee implies. We would be grateful if a more specific reference to such a study was provided. We would be happy to compare ConforFormer to a model utilizing this approach; however, we are reasonably sure that modern transformer-based models (post- Unimol) all do a full retrain to optimize the benchmark results. Section 3.1 of the paper addresses this problem in detail.
>
> > 3.	Underwhelming Performance
> As shown in Tables 1 and 2, the model's performance falls short of state-of-the-art methods across multiple benchmarks. The results do not convincingly demonstrate that the proposed approach yields meaningful improvements in molecular representation quality.
>
> The model we train for the downstream tasks is a 3-layer MLP on top of a fixed embedding. This setup is competing against fully flexible transformer-based architectures which are retrained for each individual task. In our opinion, the performance of these “frozen embeddings” is significantly better than expected (on the biggest, most challenging MUV dataset it is on par with fully unfrozen Uni-Mol), and our embeddings also are capable of distinguishing whether two arrangements of atoms belong to the same molecule or not.
>
> > Questionable Embedding Behavior
> Figure 5 presents a pair of conformers with substantial geometric differences. If ConforFormer-OMol had truly learned a robust understanding of 3D molecular structure, the cosine similarity between these embeddings should be significantly lower. This example casts doubt on the model’s ability to distinguish conformational nuances.
>
> We kindly want to reiterate that this is the *intended* behavior. We want the conformers of the same molecule to have close embeddings since they should have the same properties in all practical benchmarks (molecules are flexible and will adopt different conformations in different environments anyway); Figure 5 illustrates that.
>
> The newly uploaded revision clarifies this in the figure caption and fixes an unfortunate typo in Figure 6 where scores for Conforformer-OMol and Uni-Mol were swapped (the main text still lists the correct numbers). Please also consider Figures S1-S10 in the Supporting Information, Section E for more examples of ConforFormer correctly distinguishing between conformers and isomers.

---

> > ### Comment · Reviewer_mwTZ · 2025-11-24
> >
> > Thank you for the responses!
> >
> > For an example of pretraining and fine-tuning model: https://www.nature.com/articles/s41467-024-53751-y
> >
> > As for figure 5, I thought a 3D model that learns the 3D information should be sensitive to the conformer change. If not, then it would be essentially the same as a 2D graph or even fingerprint?

---

> ### Author Response · Authors · 2025-11-25
> **Reply to the response by Reviewer mwTZ**
>
> Thank you for the response! Please find the replies below.
>
> > For an example of pretraining and fine-tuning model: https://www.nature.com/articles/s41467-024-53751-y
>
> Thank you for the provided link!  Unfortunately, the paper you referenced (Mendez-Lucio et al.) considers exclusively molecular graphs, without any 3D geometry input at all. Moreover, while the text of the paper mentions embeddings, the actual fine-tuning process they used to report the SOTA results involves updating the weights of the full model. So, respectfully, it is not particularly relevant for conformation-aware pretraining and reusing the frozen embeddings.
>
> The Supporting information of Mendez-Lucio et al. contains some discussion of using the embeddings directly as input features for XGBoost alongside other molecular properties, but they observe much larger drop in performance on classification benchmarks compared to fine-tuning the full model than we do, so I'm again not sure if the comparison is warranted.
>
> > As for figure 5, I thought a 3D model that learns the 3D information should be sensitive to the conformer change. If not, then it would be essentially the same as a 2D graph or even fingerprint?
>
> The idea is for the model to learn molecular graph equivalent information from 3D geometries (atomic coordinates) alone and store it in a compact vector. For reference, "3D geometry" means that only x, y, z coordinates of the atoms are supplied to the model and it needs to learn its own approximation for molecular bonding. Figures 5 and 6 display "sticks" corresponding to bonds, however, this is just for the visualization. Chemical graphics software utilizes a table of atomic covalent radii to produce these images, the model does not have this knowledge supplied to it.  Understanding the flexibility of a molecule and being able to determine that two given 3D geometries can freely interconvert demonstrates a deeper level of chemical knowledge by the model than assigning different embeddings to different conformers.
>
> Indeed, we can see that with the embedding space regularized in this way, performance of the frozen embeddings on a variety of benchmarks improves, even if none of these benchmarks actually require the model to distinguish conformers and isomers.

---

### Official Review · Reviewer_W54S · 2025-10-30

**Soundness:** 2
**Presentation:** 1
**Contribution:** 1
**Rating:** 2
**Confidence:** 4

**Summary:**

This paper explores the use of advanced contrastive learning techniques to enhance molecular representation learning. By introducing the ConforFormer framework, the authors aim to develop conformation-invariant molecular embeddings that capture 3D geometric information without relying on explicit molecular graphs. Although the idea is conceptually interesting and relevant to modern chemical foundation models, the method shows limited novelty beyond existing architectures such as Uni-Mol, and several claims lack sufficient experimental or theoretical support.

**Strengths:**

Using advanced contrastive learning to improve molecular representation learning is an interesting research topic.

**Weaknesses:**

1. The presentation of the paper is poor, making it difficult to understand the research problem and motivation it aims to address. In the abstract, the authors argue that existing methods using published weights only as initialization have certain limitations.
I do not think this is a real limitation, since most approaches intentionally leverage pretrained foundation models to support various downstream tasks. In the introduction, the authors claim that real-world datasets are often too small to allow stable retraining, which is also not accurate, as many domain adaptation techniques—such as few-shot learning and data augmentation—can effectively address this issue.

2. The proposed method lacks novelty. The so-called *“new weakly supervised contrastive learning objective”* is essentially the standard contrastive loss without any additional innovation. The authors claim to propose a novel structure called **ConforFormer**, but it is architecturally identical to **Uni-Mol**, except for the added contrastive learning objective.  Also, some other paper already has used Contrastive Learning for 3D molecular representation learning, see [1]

3. Some claims in the paper lack sufficient evidence. For example, the paper mentions *“a benchmark evaluating the model's ability,”* but there is no open-source release or supporting evidence provided to describe the benchmark in detail.

[1]Qin, Jiayu, et al. "A probability contrastive learning framework for 3D molecular representation learning." Advances in Neural Information Processing Systems 37 (2024): 58058-58076.

**Questions:**

What are the detailed definitions of the loss terms in the total loss (e.g., L_token, L_coord, L_distance)?
How were these terms computed, and how were the coefficients (5, 10, 2) determined — empirically or theoretically?

---

> ### Author Response · Authors · 2025-11-24
> **Reply to Reviewer W54S**
>
> Thank you for the review! Please find the replies to your concerns below.
>
> > 1.	The presentation of the paper is poor, making it difficult to understand the research problem and motivation it aims to address. In the abstract, the authors argue that existing methods using published weights only as initialization have certain limitations.
> I do not think this is a real limitation, since most approaches intentionally leverage pretrained foundation models to support various downstream tasks. In the introduction, the authors claim that real-world datasets are often too small to allow stable retraining, which is also not accurate, as many domain adaptation techniques—such as few-shot learning and data augmentation—can effectively address this issue.
>
> We would be happy to correct the abstract if the reviewer believes it lacks clarity. In practice, real-world approaches in other domains commonly use the foundational models to generate embeddings and reuse them later.
>
> In section 3.1 we contrast these industry standard practices with full model retraining as used by the original Uni-Mol paper and show that the full-retrain approach significantly decreases the stability of the benchmark results: on BBBP and BACE datasets the standard deviation of the estimate produced by “unfrozen” Uni-Mol is an order of magnitude higher than for a frozen backbone.
>
> > 2.	The proposed method lacks novelty. The so-called “new weakly supervised contrastive learning objective” is essentially the standard contrastive loss without any additional innovation. The authors claim to propose a novel structure called ConforFormer, but it is architecturally identical to Uni-Mol, except for the added contrastive learning objective. Also, some other paper already has used Contrastive Learning for 3D molecular representation learning, see [1]
>
> Contrastive learning as a technique is well-established and is, generally, a strong method for regularizing the embedding space. We discuss this in Introduction, lines 052-059. The key innovation in these approaches lies in choosing the correct learning objective: what kind of data points should be considered similar?
>
> We are grateful for the provided reference (Jiayu Qin et al.); this is an inspiring paper, which, like ours, also uses the Uni-Mol architecture entirely unchanged except for the loss function. Jiayu Qin et al., however, focused on dynamically adjusting the positive and negative pairs in a way that different molecules (2D graphs) would be considered a positive pair in the InfoNCE loss if their properties are similar. While it would be plausible if their paper utilized the conformation space for contrastive learning, neither its text nor the supplied code mentions this in any way. Moreover, the paper’s introduction (page 2, third paragraph) explicitly states that “contrastive learning setup should consider molecules with structural similarities as positive pairs, even when they originates [sic] from different molecules”.
>
> To reiterate, the focus of our paper was to get the same representation for distinct 3D geometries if and only if they belong to the same molecule. If the reviewer is aware of publications exploring this regularization of molecular representation space, we would be happy to read them.
>
> > 3.	Some claims in the paper lack sufficient evidence. For example, the paper mentions “a benchmark evaluating the model's ability,” but there is no open-source release or supporting evidence provided to describe the benchmark in detail.
>
> This benchmark is described in detail in the main paper text, section 4.1, with technical details of constructing it referenced and linked in the main text to the section B of Supporting Information.  The benchmark is intended to be published alongside the paper, but it was not possible to do so anonymously. We submitted a sample of the benchmark data as a part of the Supporting information.
>
> > What are the detailed definitions of the loss terms in the total loss (e.g., L_token, L_coord, L_distance)?
>
> As noted in the paper text below the loss equation in Section 3.3 (lines 260-264)
>
> >"Here $L_{token}$ is the loss associated with the masked token prediction, $L_{coord}$ is the loss associated with the coordinates de-noising task, and $L_{distance}$ is the loss associated with the masked distance prediction. The batch for computing these losses consists of $n=128$ unique molecules (without additional conformers added). These were introduced in Zhou et al. (2022) and described in detail in Ji et al. (2024)."
>
> Detailed description of these losses can be found in the referenced Uni-Mol papers.
>
> > How were these terms computed, and how were the coefficients (5, 10, 2) determined — empirically or theoretically?
>
> The coefficients 1, 5, 10 were taken from the Uni-Mol code directly, coefficient 2 for the contrastive loss was chosen empirically.

---

### Official Review · Reviewer_PKKJ · 2025-11-03

**Soundness:** 2
**Presentation:** 3
**Contribution:** 2
**Rating:** 4
**Confidence:** 3

**Summary:**

This paper proposes ConforFormer, a Transformer-based molecular representation model that aims to learn conformation-invariant molecular embeddings through contrastive learning across different 3D conformers of the same molecule. The goal is to obtain general molecular representations that capture structural consistency without requiring task-specific fine-tuning.

**Strengths:**

1. The paper targets a meaningful and relevant problem, how to build robust molecular representations that account for 3D conformational variability.
2. The proposed framework is conceptually clear and easy to follow, with a reasonable motivation and solid experimental setup.
3. The introduction of the PharmIsomer benchmark provides an interesting way to evaluate whether models can distinguish between conformers and isomers. The writing and figures are clear, making the overall presentation accessible.

**Weaknesses:**

1. Limited technical novelty: The approach mainly extends existing ideas from contrastive learning and 3D molecular representation (e.g., Uni-Mol) without introducing substantial methodological innovation.
2. The results do not show clear or consistent improvements over strong baselines such as Uni-Mol; in some benchmarks, performance is even slightly worse. This weakens the paper’s contribution, since if training the baseline is not computationally expensive, practitioners would still prefer to fine-tune an existing model rather than use ConforFormer’s frozen representation.

**Questions:**

Please refer to the cons

---

> ### Author Response · Authors · 2025-11-24
> **Reply to Reviewer PKKJ**
>
> Thank you for the review! Please find the replies to your concerns below.
>
> > 1.	Limited technical novelty: The approach mainly extends existing ideas from contrastive learning and 3D molecular representation (e.g., Uni-Mol) without introducing substantial methodological innovation.
>
> To the best of our knowledge, there are no literature examples of a model trained exclusively on 3D structures becoming able to distinguish structures with distinct molecular graphs. Contrastive learning is a powerful general technique, with the result of applying it mostly determined by the choice of positive and negative pairs. Here, the novelty is in utilizing it to ensure that the model learns to infer the molecular graph from the 3D representation without 2D structure being explicitly provided.
>
> The additional methodological innovation of the current work lies in building a new benchmark dataset for the ability of models to distinguish isomers from their 3D structures: this is a previously unexplored property of molecular representation, which we believe is key for reliable modeling of molecular properties.
>
>
> > 2.The results do not show clear or consistent improvements over strong baselines such as Uni-Mol; in some benchmarks, performance is even slightly worse. This weakens the paper’s contribution, since if training the baseline is not computationally expensive, practitioners would still prefer to fine-tune an existing model rather than use ConforFormer’s frozen representation.
>
> We kindly disagree with the referee, as we show a consistent improvement over the frozen representation obtainable from Uni-Mol. We are aware that fully flexible models achieve better scores on most of the benchmarks (notably excluding the biggest biochemical benchmark in this study, MUV). However, this does not disqualify three key findings of the paper:
> 1) that Uni-Mol-like models can produce useful frozen embeddings for molecular representation;
> 2) that this embedding space can be regularized by contrastive learning where conformers of the same molecule are considered a positive pair;
> 3) that this simple additional objective leads to emergent capability of the model to distinguish molecular graphs from 3D geometries alone, without being directly trained for this.
>
> To quote the paper text regarding the improvements,
>
> *“Training on this sub-dataset (further OMol) without contrastive loss produced results on par with Uni-Mol with no flat structures, but training with contrastive loss and freezing the model produced the embeddings of higher quality, performing the best on 4 out of 6 quantum-chemical benchmarks, and tying for the first place in 5 out of 8 classification benchmarks”*

---

### Author Response · Authors · 2025-11-24

Minor changes (updated wording in the abstract, update to Figure 5 and 6 captions) as detailed in the replies to individual reviewers.

---

### Meta-Review · Area_Chair_fB5M · 2025-12-28

**Summary:**

1. **Limited Novelty**:

   * A significant concern was that the **novelty** of the proposed **ConforFormer** model is limited. The paper was seen as an incremental extension of existing works like **Uni-Mol** with the addition of contrastive learning and some minor modifications (e.g., removing "flat" structures from training data). Reviewers noted that **contrastive learning** has already been applied in other works, and the methodology didn’t present substantial new innovations. This lack of a novel approach was emphasized as a critical weakness.

2. **Misrepresentation of Prior Work**:

   * Some reviewers felt that the paper misrepresented existing works in the field. Specifically, the claim that **no chemical embedding model captures the diversity of 3D molecular conformations** was considered misleading, as there are existing models explicitly addressing 3D conformational diversity. This raised concerns about the scholarly rigor of the paper and its engagement with prior literature.

3. **Underwhelming Performance**:

   * Several reviewers pointed out that the model’s performance fell short of **state-of-the-art** methods. The results from **ConforFormer** were not convincingly better than the **Uni-Mol** baseline, and in some cases, the model performed **worse**. This undermined the paper's claims of improvement and made it difficult to justify the use of **frozen embeddings** as the core advantage, especially when fine-tuning existing models might yield better results.

4. **Unclear Contrastive Learning Objective**:

   * The reviewers were unclear about the differences between the proposed **contrastive loss** and the existing **Uni-Mol** contrastive learning approach. They requested a more detailed explanation of how the new objective differs from previously used methods.

5. **Dataset Issues and Evaluation**:

   * There were concerns about the **datasets** used, particularly the **OMol** dataset, which did not show significant improvements over the **Uni-Mol** dataset. The effectiveness of **3D conformations** was questioned, as there was no baseline comparison using only **graph data** (without 3D information) to assess the true value of incorporating **3D geometry** into the model. Furthermore, some claimed that **RDKit-generated 3D coordinates** could degrade performance.

6. **Terminology and Clarity Issues**:

   * The reviewers pointed out that some **terminology** was used incorrectly, such as the term “**retrain**” when **fine-tune** was meant. The confusion around terminology weakened the clarity of the manuscript and raised concerns about the authors' understanding of **machine learning concepts**. Additionally, the use of the term **task-agnostic** was seen as an **overstatement**, given that the authors still fine-tune the model for specific downstream tasks.

7. **Presentation and Clarity**:

   * The presentation of the paper was criticized for being unclear and difficult to follow. Some reviewers felt that the **research problem** was not well-explained, and the **motivation** behind the study was weak. For example, the paper argued that existing approaches with pretrained weights have limitations, but the reviewers felt this claim was not sufficiently supported. Similarly, the arguments around the inability to "retrain" existing models were not convincing to some reviewers.

8. **Evaluation of Conformational Understanding**:

   * A specific example raised by reviewers was **Figure 5**, where **ConforFormer** showed high similarity in the embeddings of two **conformers** with substantial geometric differences. This led to doubts about whether the model truly understands **3D molecular structure**. The reviewers questioned if the model was effectively learning the **flexibility** of molecules, as the embeddings for highly different conformers should ideally be more distinct.

**Reviewer Concerns:**

### **Reviewer Concerns - Addressed and Unaddressed**

#### **Unaddressed Concerns:**

1. **Novelty of the Approach**:
   Despite the authors' claim that their method introduces a **novel contrastive learning objective** for 3D molecular embeddings, the reviewers remained unconvinced that the **innovation** goes beyond **existing methods** like **Uni-Mol**. The rebuttal emphasizes the importance of **frozen embeddings** and **contrastive loss** but fails to sufficiently distinguish this work from similar methods already explored in the field. This concern about **limited novelty** remains unaddressed.

2. **Underwhelming Performance**:
   While the authors argue that **frozen embeddings** provide a practical alternative to full model retraining, the reviewers were still concerned that **ConforFormer** did not outperform existing models, and in some cases, performed **worse**. The lack of clear **improvements** on several benchmarks raises doubts about the method’s contribution to advancing molecular representation learning.

3. **Misrepresentation of Prior Work**:
   The reviewers noted that the authors’ claim—“no chemical embedding model capturing the diversity of 3D molecular conformations”—overlooks significant prior work on **conformer-aware pretraining**. Although the authors provided some clarification, the reviewers did not find the rebuttal sufficiently compelling to change their view on the **lack of engagement with existing literature**.

4. **Evaluation and Dataset Issues**:
   The concerns regarding the **OMol dataset** and the absence of a baseline using **graph data (without 3D information)** were not sufficiently resolved in the rebuttal. The authors did not provide a convincing argument for why their approach is more effective than simply using 2D representations or **RDKit-generated 3D coordinates**, which some reviewers believe can even degrade performance.

#### **Addressed Concerns:**

1. **Terminology and Clarity**:
   The authors clarified the misuse of the terms "**retrain**" versus "**fine-tune**" and updated the abstract and main text accordingly. The response also addressed the confusion about the term **task-agnostic**, and the authors acknowledged the use of **frozen embeddings** and their advantage for small datasets.

2. **Contrastive Learning Objective**:
   The authors provided a more detailed explanation of the **contrastive loss** and emphasized that it was not part of **Uni-Mol**, addressing the reviewer's concern about the lack of novelty. The clarification regarding how the method distinguishes conformers of the same molecule through **contrastive learning** was partly effective in addressing some concerns, although reviewers still questioned whether it was enough to justify the novelty.

3. **Evaluation of Conformational Understanding**:
   The authors defended the **embedding similarity** observed in **Figure 5**, explaining that the model is designed to treat different conformers of the same molecule as the same entity, which is consistent with its goal. The reviewers’ concerns about the model’s ability to distinguish conformational nuances were partly addressed, although they remained unconvinced about the model’s true capability.

**Reviewer Scores:**

#### **Reviewer 1**

* **Original Score**: 6 (Marginally Above the Acceptance Threshold)

* **Reviewer’s Concerns**:
  Reviewer raised concerns regarding **novelty**, the **lack of improvement** over baselines, and the unclear **contribution** to existing methods like **Uni-Mol**.

* **Likelihood of Score Change**:
  Given the clarification in the rebuttal regarding **contrastive learning** and **frozen embeddings**, the reviewer might have acknowledged some improvements but still remained unconvinced about the overall **novelty** and the **performance** improvements over the baseline. Despite the additional explanation of the method’s intent to model **molecular flexibility**, the lack of substantial evidence to support significant improvements on benchmarks might lead the reviewer to maintain their score at **6**.

#### **Reviewer 2**

* **Original Score**: 2 (Reject)

* **Reviewer’s Concerns**:
  Reviewer highlighted **limited novelty**, underwhelming **performance**, and the **misrepresentation of prior work** in the field.

* **Likelihood of Score Change**:
  The reviewer’s concerns regarding **performance** and **novelty** were not fully addressed, and they remained skeptical about the method’s actual contributions. While the rebuttal did clarify the authors' stance on **contrastive learning**, the reviewer’s doubts about the **effectiveness** of the approach and the **engagement with prior work** likely would lead them to maintain their **reject** decision. The score would likely remain at **2 (Reject)**.

#### **Reviewer 3**

* **Original Score**: 4 (Marginally Below the Acceptance Threshold)

* **Reviewer’s Concerns**:
  Reviewer raised issues about **novelty**, **benchmark results**, and the **effectiveness of 3D geometry** representations.

* **Likelihood of Score Change**:
  Despite the authors’ clarifications on **contrastive learning** and how their model addresses **conformational variations**, the reviewer was still unconvinced about the **novelty** of the approach and its **performance**. They remained skeptical about whether the **3D geometry** representation provided significant advantages over **graph-based approaches**. Given these concerns, the reviewer would likely **not** change their score and would remain at **4 (Marginally Below the Acceptance Threshold)**.

#### **Reviewer 4**

* **Original Score**: 2 (Reject)

* **Reviewer’s Concerns**:
  Reviewer emphasized that the paper reads like a **technical report** with **incremental modifications** to **Uni-Mol** and **limited novelty** in the methodology.

* **Likelihood of Score Change**:
  The rebuttal did clarify several aspects, including the **contrastive learning objective** and how the **frozen embeddings** are utilized, but the reviewer was still unconvinced that the paper provided substantial **novelty** or **impact** over existing methods. Given that the rebuttal did not significantly address the **core concerns** of **incremental contribution** and **underwhelming novelty**, the reviewer would likely **not** change their score, keeping it at **2 (Reject)**.

---

### Decision · Program_Chairs · 2026-01-26

Reject